# SPELL: Self-Play Reinforcement Learning for Evolving Long-Context Language Models

**Ziyi Yang**[1,2*]   **Weizhou Shen**[2]   **Chenliang Li**[2]   **Ruijun Chen**[1]   **Fanqi Wan**[1]
**Ming Yan**[2†]   **Xiaojun Quan**[1,3†]   **Fei Huang**[2]
[1]Sun Yat-sen University   [2]Tongyi Lab, Alibaba Group   [3]Shenzhen Loop Area Institute
`yangzy39@mail2.sysu.edu.cn, ym119608@alibaba-inc.com`
`xiaojunquan@slai.edu.cn`

## Abstract

Progress in long-context reasoning for large language models (LLMs) has lagged behind other recent advances. This gap arises not only from the intrinsic difficulty of processing long texts, but also from the scarcity of reliable human annotations and programmatically verifiable reward signals. In this paper, we propose **SPELL**, a multi-role self-play reinforcement learning framework that enables scalable, label-free optimization for long-context reasoning. SPELL integrates three cyclical roles—*questioner*, *responder*, and *verifier*—within a single model to enable continual self-improvement. The questioner generates questions from raw documents paired with reference answers; the responder learns to solve these questions based on the documents; and the verifier evaluates semantic equivalence between the responder's output and the questioner's reference answer, producing reward signals to guide continual training. To stabilize training, we introduce an automated curriculum that gradually increases document length and a reward function that adapts question difficulty to the model's evolving capabilities. Extensive experiments on six long-context benchmarks show that SPELL consistently improves performance across diverse LLMs and outperforms equally sized models fine-tuned on large-scale annotated data. Notably, SPELL achieves an average 7.6-point gain in pass@8 on the strong reasoning model Qwen3-30B-A3B-Thinking, raising its performance ceiling and showing promise for scaling to even more capable models. Our code is available at `https://github.com/Tongyi-Zhiwen/Qwen-Doc`.

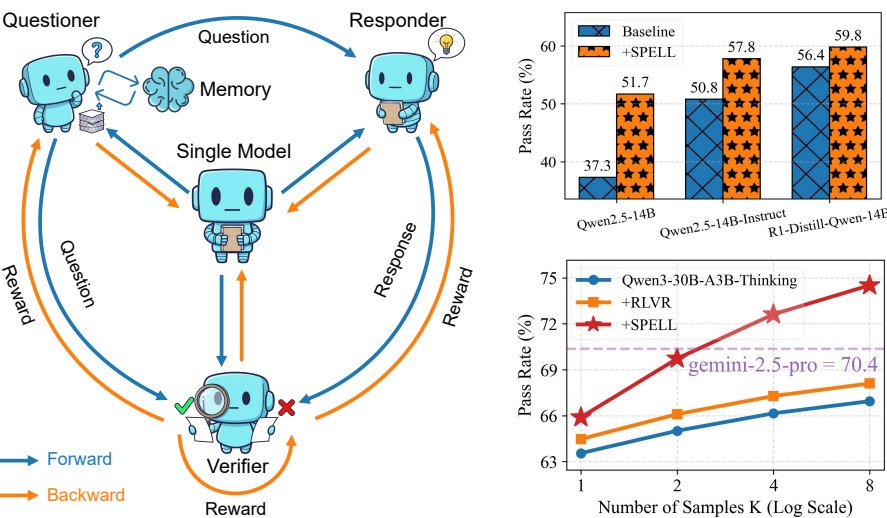

Figure 1: **(Left)** An overview of the SPELL framework, where a single LLM self-evolves by dynamically adopting the roles of *questioner*, *responder*, and *verifier*. **(Right)** SPELL consistently boosts performance across various models (*top*) and exhibits superior test-time scaling over traditional RLVR (*bottom*).

* Work done during internship at Tongyi Lab, Alibaba Group.
† Corresponding authors.

## 1 INTRODUCTION

In recent years, reinforcement learning (RL) has emerged as a promising approach for enhancing the reasoning capabilities of large language models (LLMs) (Guo et al., 2025; Yang et al., 2025; Jaech et al., 2024; Team et al., 2025). Among these methods, reinforcement learning with verifiable rewards (RLVR) has shown particular promise in domains where correctness can be programmatically verified, such as mathematics, logical reasoning, and software engineering (Lambert et al., 2024; Hu et al., 2025; Liu et al., 2025c; Wei et al., 2025). RLVR methods employ rule-based or programmatic verifiers to generate reward signals, which then guide policy optimization through algorithms such as Proximal Policy Optimization (PPO) (Schulman et al., 2017b), Group Relative Policy Optimization (GRPO) (Shao et al., 2024), and related variants (Shao et al., 2024; Yue et al., 2025; Liu et al., 2025d).

Despite these advances, most RLVR research has been restricted to short-context settings (e.g., <1024 tokens), where models primarily rely on their parametric knowledge for reasoning (Wan et al., 2025). In contrast, reasoning over long documents like long-context question answering requires not only locating relevant evidence scattered across extended contexts but also executing multi-step reasoning. Extending RLVR to long-context reasoning presents significant challenges, which stem from the inherent difficulty of processing long texts, as well as two critical bottlenecks: the prohibitive cost and unreliability of human annotations, and the absence of programmatically verifiable rewards.

Empirical evidence highlights the severity of these issues. On benchmarks such as LongBench-V2, human accuracy for extra-long multiple-choice reasoning tasks drops to 25.1% — effectively approaching random chance (Bai et al., 2025). This not only limits the performance achievable under human supervision but also imposes a scalability ceiling, particularly as LLMs approach superhuman reasoning capabilities (Zhao et al., 2025). Specifically, as context length grows, producing reliable annotations becomes increasingly costly and unstable, and supervision diversity diminishes. Moreover, the lack of verifiable reward mechanisms in long-context settings further constrains the applicability of RLVR, posing a fundamental challenge to advancing reasoning capabilities at scale.

To address these limitations, we turn to self-play RL, where a single model learns to self-evolve by generating and solving its own tasks without human labels (Zhou et al., 2025; Chen et al., 2025b; Huang et al., 2025). However, applying self-play to long-context reasoning poses a unique challenge: answers may be semantically correct yet differ substantially in expression, rendering string matching or naive majority voting unreliable reward signals. Thus, the model should not only generate questions and answers, but also verify its own solutions reliably. This observation motivates our framework, in which one LLM assumes three complementary roles: *questioning*, *responding*, and *verifying*.

In this paper, we introduce **SPELL** (**S**elf-**P**lay Reinforcement Learning for **E**volving **L**ong-Context **L**anguage Models), a self-play RL framework for long-context reasoning. In this setup, a unified policy alternates among three roles: the *questioner*, which formulates questions with reference answers from raw documents; the *responder*, which attempts to solve them; and the *verifier*, which compares the responder's output with the reference answer to produce reward signals for joint optimization. To steer this process, SPELL incorporates three key design elements. First, a verifier trained for self-consistency on verifiable tasks produces stable rewards, even for outputs that cannot be verified by strict rules, thereby overcoming the brittleness of string matching. Second, an automated curriculum uses a history memory of question–answer pairs and documents to progressively increase task difficulty. A Gaussian-shaped reward further calibrates difficulty around the responder's competence frontier, ensuring questions are neither too easy nor impossibly difficult. Third, a role-specific dynamic sampling strategy balances contributions across roles to stabilize training of the shared policy. Together, these components form a self-sufficient, closed-loop system that enables LLMs to autonomously evolve long-context reasoning without human-labeled data

We evaluate SPELL across 12 open-source LLMs ranging from 4B to 32B parameters, including both dense and Mixture-of-Experts (MoE) architectures. On six long-context QA benchmarks, SPELL delivers consistent performance gains. Remarkably, training a base model with SPELL enables it to surpass its instruction-tuned counterpart that relies on extensive human-annotated data, highlighting the data efficiency of our label-free self-play approach. Against a strong RLVR baseline trained on a static dataset synthesized by DeepSeek-R1-0528 (Guo et al., 2025), SPELL achieves larger and more reliable gains. For capable models such as Qwen3-30B-A3B-Thinking, SPELL's dynamic curriculum continually elevates performance and enables it to outperform the leading gemini-2.5-pro (Comanici et al., 2025) in pass@4. These findings firmly establish our self-play approach as a scalable and effective path toward advanced long-context reasoning without human supervision.

## 2 PRELIMINARIES

**Long-Context Reinforcement Learning**  We formulate the long-context generation task as a reinforcement learning (RL) problem. Given a set of $n$ documents $\{c_i\}_{i=1}^n$ and a question $q$, the goal of long-context RL is to optimize a policy model $\pi_\theta$ to generate a response $y$ that maximizes a reward function $r_\phi(c, q, y)$. The standard objective is to maximize the KL-regularized expected reward (Schulman et al., 2017a; Wan et al., 2025):

$$\max_{\pi_\theta} \mathbb{E}_{c,q\sim\mathcal{D}, y\sim\pi_\theta(\cdot|c,q)} \left[r_\phi(c, q, y)\right] - \beta\mathbb{D}_{\text{KL}}\left[\pi_\theta(y \mid c, q) \,||\, \pi_{\text{ref}}(y \mid c, q)\right], \quad (1)$$

where $c = \text{Concat}(c_1, c_2, \ldots, c_n)$, $\mathcal{D}$ is the training dataset, $\pi_{\text{ref}}$ denotes a reference policy, and $\beta$ controls the strength of the KL regularization to prevent large deviations from the reference policy.

**Group Relative Policy Optimization (GRPO)**  For long-context inputs, the quadratic complexity of the attention mechanism renders PPO (Schulman et al., 2017b), which relies on generalized advantage estimation (GAE) (Schulman et al., 2015) via a value network, computationally prohibitive. Therefore, we employ GRPO (Shao et al., 2024) to optimize the objective in Eq. (1). For each input $(c, q)$, GRPO first samples a group of $G$ candidate responses $\{y_i\}_{i=1}^G$ from the old policy $\pi_{\theta_{\text{old}}}$. It then estimates the advantage through group-wise reward $z$-score normalization, thereby obviating the need for a separate value network. Formally, the objective is:

$$\mathcal{J}_{\text{GRPO}}(\theta) = \mathbb{E}_{c,q\sim\mathcal{D}, \{y_i\}_{i=1}^G \sim\pi_{\theta_{\text{old}}}(\cdot|c,q)} \left[ \frac{1}{G}\sum_{i=1}^G \frac{1}{|y_i|}\sum_{t=1}^{|y_i|} \left( \min\Big(\rho_{i,t}(\theta)A_i, \right.\right.$$

$$\left.\left. \text{clip}\Big(\rho_{i,t}(\theta), 1-\varepsilon, 1+\varepsilon\Big)A_i\right) - \beta\mathbb{D}_{\text{KL}}(\pi_\theta||\pi_{\text{ref}})\right)\right], \quad (2)$$

where $\rho_{i,t}(\theta) = \frac{\pi_\theta(y_{i,t}|c,q,y_{i,<t})}{\pi_{\theta_{\text{old}}}(y_{i,t}|c,q,y_{i,<t})}$ is the importance sampling ratio for token $t$ in sequence $i$. The group-relative advantage $A_i$ is shared across tokens of the $i$-th sequence and computed by normalizing the sequence-level rewards $\{r_i\}_{i=1}^G$:

$$A_i = \frac{r_i - \text{mean}(\{r_k\}_{k=1}^G)}{\text{std}(\{r_k\}_{k=1}^G)}. \quad (3)$$

## 3 THE SPELL FRAMEWORK

In this section, we detail the core design of SPELL, a self-play reinforcement learning framework that enables LLMs to improve their long-context reasoning capabilities without external supervision. The key principle of SPELL is that a single policy model $\pi_\theta$ dynamically assumes three complementary roles: a *questioner* $\pi_\theta^{\text{que}}$, a *responder* $\pi_\theta^{\text{res}}$, and a *verifier* $\pi_\theta^{\text{ver}}$. Through their interaction, the model autonomously generates and solves questions while producing reliable reward signals. This closed-loop interaction creates an evolving curriculum in which the model progressively adapts to longer contexts and more complex reasoning (Section 3.1). Role-specific reward designs (Section 3.2) and a unified optimization procedure (Section 3.3) jointly drive this co-evolution.

### 3.1 THE SELF-PLAY EVOLUTIONARY LOOP

As illustrated in Figure 2 and Algorithm 1, SPELL proceeds iteratively: given a cluster of $n$ documents $C = \{c_i\}_{i=1}^n$ and a task type[1] $\tau$, the policy $\pi_\theta$ first generates new questions,[2] then attempts to solve them, and finally verifies the solutions before performing a unified policy update.

**Questioning**  The questioner $\pi_\theta^{\text{que}}$ generates new question–answer pairs in an iterative curriculum. In the very first iteration, it is conditioned only on a randomly sampled subset of $m$ documents ($m < n$) and produces a pair $(q, a)$. After each solvable pair is created, we append it to a *history memory* $\mathcal{H}$ that stores the $L$ most recent solvable question–answer pairs and their associated source documents: $\mathcal{H}_C = \{(C_l, q_l, a_l)\}_{l=1}^L$. In subsequent iterations, the questioner is conditioned on both a newly sampled

---

[1]Details of dataset construction and task definition are provided in Appendix E.1.

[2]To direct the policy in enacting three distinct roles, we adopt zero-shot prompting using tailored templates for each role and task type. Details of these templates are provided in Appendix G.

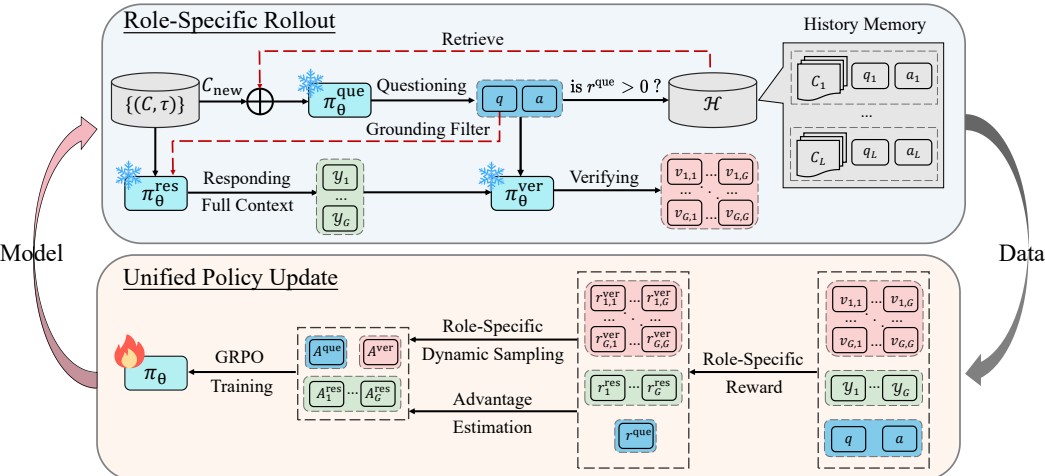

Figure 2: Overview of our proposed SPELL for self-evolution of long-context reasoning. The process operates in a continuous loop that alternates between two stages: (1) **Role-Specific Rollout**, where a single policy model enacts three distinct roles—a *questioner* ($\pi_\theta^{\text{que}}$), a *responder* ($\pi_\theta^{\text{res}}$), and a *verifier* ($\pi_\theta^{\text{ver}}$)—to generate training data. (2) **Unified Policy Update**, where the unified policy is refined using the collected data, and the enhanced model serves as the starting point for the next rollout cycle.

subset $C_{\text{new}}$ and the stored memory. The resulting context is $X^{\text{que}} = (\bigcup_{l=1}^{L} C_l) \cup C_{\text{new}} \cup \{(q_l, a_l)\}_{l=1}^{L}$. As the memory fills, the context for the questioner expands to include both previously seen and newly sampled documents, which allows the questioner to generate questions that integrate information across more documents. The history memory also raises difficulty by including past $\{(q_l, a_l)\}$: these exemplars discourage redundancy and, via prompting, push $\pi_\theta^{\text{que}}$ to generate harder questions than those already solved. Consequently, the questioner's difficulty increases for two complementary reasons: (1) the context $X^{\text{que}}$ expands over iterations as more documents are brought into scope, and (2) explicit conditioning on historical $\{(q_l, a_l)\}$ encourages the model to escalate question complexity.

**Responding** The responder $\pi_\theta^{\text{res}}$ attempts to solve the generated question based on documents. To mitigate the generation of non-grounded or hallucinated questions, we employ a grounding filter process to discard questions that can be answered without documents. For valid questions, the responder is presented with the complete set of $n$ documents, where the remaining documents unseen by the questioner serve as distractors to increase grounding and reasoning difficulty. This design enforces reliance on the provided document context rather than parametric memory. To encourage exploration of diverse reasoning trajectories, the responder generates $G$ independent rollouts $\{y_i\}_{i=1}^{G}$.

**Verifying** The verifier $\pi_\theta^{\text{ver}}$ evaluates the semantic equivalence between the responder's output $y_i$ and the questioner's reference answer $a$. For each $y_i$, it produces $G$ independent binary judgments $\{v_{i,j}\}_{j=1}^{G}, v_{i,j} \in \{0, 1\}$, which are then aggregated through majority voting:

$$v_i^{\text{ver}} = \mathbb{I}\left(\sum_{j=1}^{G} v_{i,j} > \frac{G}{2}\right), \tag{4}$$

where $\mathbb{I}(\cdot)$ is the indicator function. This ensemble-based verification reduces variance and produces a stable, semantically aware reward signal, which is essential for sustaining a self-play system.

## 3.2 ROLE-SPECIFIC REWARD DESIGN

The three roles co-evolve under specialized rewards that align their objectives while remaining compatible within a single shared policy. In what follows, we detail these rewards.

**Verifier** The verifier is trained to improve its judgment reliability through self-consistency (Wang et al., 2022; Zuo et al., 2025). For a candidate output $y_i$, the verifier produces $G$ rollouts with judgments $v_{i,j}$. Each rollout is then assigned a reward:

$$r_{i,j}^{\text{ver}} = \mathbb{I}(v_{i,j} = v_i^{\text{ver}}), \tag{5}$$

where $v_i^{\text{ver}}$ is the majority vote over $G$ rollouts.

**Responder**  The responder's reward for the $i$-th solution is the maximum of a deterministic, rule-based check and the verifier's consensus score, denoted as:

$$r_i^{\text{res}} = \max\left(\mathcal{R}_{\text{rule}}(y_i, a), v_i^{\text{ver}}\right). \tag{6}$$

The rule-based function, $\mathcal{R}_{\text{rule}}$, provides a binary reward based on cover exact match (CEM) criteria (Wan et al., 2025; Song et al., 2025)—it returns 1 if the ground-truth answer $a$ appears in the generated response $y_i$ and 0 otherwise. The maximum reward plays a crucial role: when $y_i$ is a correct paraphrase that CEM fails to capture, a majority vote of $v_i^{\text{ver}} = 1$ prevents the policy from being misled by false-negative noise, which stabilizes learning and encourages continual improvement.

**Questioner**  The questioner is incentivized to generate questions of intermediate difficulty, as learning is most efficient at the frontier of the LLM's capabilities (Bae et al., 2025; Huang et al., 2025). For binary-reward tasks, this frontier corresponds to a success probability of 0.5, which maximizes reward variance and provides the richest learning signal. We therefore define the questioner's reward as a Gaussian function centered at this optimal point. Given the responder's average success rate, $\bar{r}^{\text{res}} = \frac{1}{G} \sum_{i=1}^{G} r_i^{\text{res}}$, the reward is:

$$r^{\text{que}} = \begin{cases} \exp\left(-\frac{(\bar{r}^{\text{res}} - \mu)^2}{2\sigma^2}\right) & \text{if } 0 < \bar{r}^{\text{res}} < 1 \\ 0 & \text{if } \bar{r}^{\text{res}} = 0 \text{ or } \bar{r}^{\text{res}} = 1 \\ -0.5 & \text{if the question is not grounded in documents} \\ -1 & \text{if the question-answer pair has formatting errors} \end{cases} \tag{7}$$

We set the mean $\mu = 0.5$ to target the point of maximum learning efficiency and the standard deviation $\sigma = 0.5/3$ to concentrate the reward around this level. Additionally, the questioner is penalized for producing ill-formatted (e.g., non-parsable) question–answer pairs or questions that can be solved without context, thereby enforcing both correct formatting and strong grounding in the provided text.

## 3.3 Unified Policy Optimization

A central feature of SPELL is that samples generated under different roles supervise a single policy $\pi_\theta$. The optimization must control both sample efficiency and gradient balance across roles.

**Role-Specific Dynamic Sampling**  The raw samples collected for each document instance are highly imbalanced: one questioner sample, $G$ responder samples, and $G^2$ verifier judgments. To prevent the verifier's samples from dominating updates and to prioritize improvements in the responder's document-grounded reasoning, we introduce a role-specific sampling strategy that leverages the statistical structure of each role's signals. For the responder, we retain all groups with non-zero reward variance ($\text{std}(\{r_i^{\text{res}}\}_{i=1}^{G}) > 0$). The associated questions are labeled as positives for the questioner, and an equal number of negatives are drawn from questions with non-positive reward, as defined in Eq. (7). For the verifier, we preserve instances where the majority vote agrees with the rule-based check and subsample groups with conflicting verifications to match the number of questions. This role-specific sampling strategy reduces the training set to roughly $1/G$ of all samples, accelerates optimization, and prevents the responder's gradients from being overwhelmed by verifier samples. Importantly, although most verifier samples are omitted, their collection cost is low, see Appendix F.1.

**Advantage Estimation**  For the responder and verifier, which generate $G$ outputs per prompt, we use group-level advantage estimation as defined in Eq. (3):

$$A_i^{\text{role}} = \frac{r_i^{\text{role}} - \text{mean}(\{r_k^{\text{role}}\}_{k=1}^{G})}{\text{std}(\{r_k^{\text{role}}\}_{k=1}^{G})}, \ \text{role} \in \{\text{res}, \text{ver}\}. \tag{8}$$

The questioner generates only a single output per instance and thus lacks a group-level baseline. Therefore, we adapt the normalization method from REINFORCE++-baseline (Hu, 2025) and normalize its reward against other questioner rewards within the training batch $\mathcal{B}^{\text{que}}$:

$$A^{\text{que}} = \frac{r^{\text{que}} - \text{mean}(r^{\text{que}} \mid r^{\text{que}} \in \mathcal{B}^{\text{que}})}{\text{std}(r^{\text{que}} \mid r^{\text{que}} \in \mathcal{B}^{\text{que}})}. \tag{9}$$

**Unified Policy Update**  After collecting and sampling a batch of samples, the policy parameters $\theta$ are updated by jointly optimizing the GRPO objective across all three roles:

$$\mathcal{J}_{\text{GRPO}}(\theta) = \mathcal{J}_{\text{GRPO}}^{\text{que}}(\theta) + \mathcal{J}_{\text{GRPO}}^{\text{res}}(\theta) + \mathcal{J}_{\text{GRPO}}^{\text{ver}}(\theta) \tag{10}$$

The updated $\pi_\theta$ is reused to execute all roles in the next iteration. This closes the self-evolutionary cycle and keeps one unified policy for questioning, responding, and verifying.

## 4 EXPERIMENTS

### 4.1 EXPERIMENTAL SETUP

**Training Details**   Our SPELL RL framework is implemented using VeRL (Sheng et al., 2025). During generation, we employ a sampling temperature of 0.7 and a top-$p$ value of 0.95. The maximum input length is 16K tokens, while the maximum output length is set to 4K for non-reasoning models and extended to 20K tokens for reasoning models. To balance rollout diversity and computational efficiency, we utilize a group size of $G = 8$. The maximum number of recent solvable question–answer pairs cached in history memory is set to $L = 3$, and the number of candidate documents drawn when proposing a new question is set to $m = 5$. We conduct a purely on-policy RL training with a batch size of 128 and a constant learning rate of $2 \times 10^{-6}$. At the beginning of each rollout, we randomly sample one of three predefined task formats—document general QA, financial math QA, or multiple-choice—along with a relevant document list from the corpus. Prompt templates for each task $\tau$ and each role are provided in Appendix G. For the RLVR baseline, we synthesize a dataset using DeepSeek-R1-0528 (Guo et al., 2025) over the same document corpus and maintain identical hyperparameters to ensure a fair comparison. For comprehensive details on data construction, RL algorithm, and baselines, please refer to Appendix E.1, E.3, and E.5.

**Evaluation Benchmarks**   We evaluate our models on six long-context benchmarks, spanning multiple-choice QA on LongBench-V2 (Bai et al., 2025) and multi-hop QA across Frames (Krishna et al., 2025), HotpotQA (Yang et al., 2018), 2WikiMultihopQA (Ho et al., 2020), MuSiQue (Trivedi et al., 2022)[3], and the DocMath (Zhao et al., 2024) for financial report reasoning task. We evaluate all models with maximum input lengths of 16K and 100K tokens, and report the average accuracy over eight runs. Further details on the benchmarks and evaluation protocol are available in Appendix E.2.

### 4.2 MAIN RESULTS

Table 1 summarizes the results of SPELL across 12 open-source LLMs on six long-context QA benchmarks under maximum input lengths of 16K and 100K tokens. These results offer valuable insights into SPELL's effectiveness and generalization, as elaborated below.

**SPELL consistently enhances performance across diverse models.**   Our self-play framework exhibits strong universality, and it delivers substantial improvements across different architectures, sizes, and families. This versatility is evident across the following dimensions. (1) **Model types and sizes**: SPELL cultivates complex reasoning skills from scratch. For unaligned base models, the average improvement at 16K is large and robust, with Qwen2.5-7B, Qwen2.5-14B, and Qwen2.5-32B improving by 13.9, 14.4, and 9.1 points, respectively. Remarkably, these trained models consistently outperform their instruction-tuned counterparts of the same size, which are trained with extensive human-annotated data. This result highlights that SPELL is data-efficient and practically valuable in scenarios where labeled data is scarce. SPELL also benefits instruction-tuned models, e.g., Qwen2.5-7B-Instruct improves by 9.0 points. For highly specialized reasoning models such as R1-Distill-Qwen-14B, the performance still increases by 3.4 points. (2) **Architecture**: Beyond dense models, the framework is also applicable to Mixture-of-Experts (MoE) models, where it improves Qwen3-30B-A3B-Instruct and Qwen3-30B-A3B-Thinking by 4.4 and 2.0 points, respectively. (3) **Model families**: Improvements extend across families. For example, Llama-3.1-8B-Instruct and R1-Distill-Llama-8B increase by 4.4 and 3.4 points, respectively. Collectively, these results establish SPELL as a broadly effective paradigm for advancing LLMs in long-context tasks.

**SPELL is superior to traditional RL with static data.**   We compare SPELL against the RLVR baseline trained on a fixed dataset synthesized by DeepSeek-R1-0528. Although such static data offers high-quality supervision for RL training, it cannot adapt to the policy's evolving capabilities. In contrast, SPELL constructs a self-play curriculum that tracks the model's current ability: the questioner focuses on instances near the responder's competence boundary, maintaining alignment between the training signal and the policy throughout optimization. The advantage becomes increasingly evident as the policy model's capabilities grow. For Qwen2.5-7B, RLVR achieves performance comparable to SPELL, indicating that a static corpus appears sufficient for weaker policies. However, for Qwen3-30B-A3B-Thinking, SPELL improves average scores by 2.0, whereas RLVR yields no gain. On the more challenging benchmarks for the same model, RLVR decreases accuracy on

---

[3]We use the subsets of HotpotQA, 2WikiMultihopQA, and MuSiQue from LongBench (Bai et al., 2024b).

Table 1: Overall results of our proposed SPELL method with maximum input lengths of 16K and 100K on long-context benchmarks. "LB-MQA" represents the average performance across 2WikiMultihopQA, HotpotQA, and MuSiQue. "LB-V2" refers to LongBench-v2. For the average score (Avg.), + indicates the relative improvement over the base model within each group. The best score in each model group is highlighted in **bold**.

| Models | 16K | | | | | 100K | | | | |
|---|---|---|---|---|---|---|---|---|---|---|
| | DocMath | Frames | LB-MQA | LB-V2 | Avg. | DocMath | Frames | LB-MQA | LB-V2 | Avg. |
| *Base Models* | | | | | | | | | | |
| Qwen2.5-7B | 10.9 | 27.9 | 36.7 | 31.2 | 26.7 | 16.1 | 24.2 | 31.2 | 22.7 | 23.6 |
| + RLVR | **41.8** | **41.0** | 50.0 | 30.2 | **40.8**$^{+14.1}$ | **42.7** | **40.3** | 49.2 | 26.0 | **39.6**$^{+16.0}$ |
| + SPELL | 40.0 | 39.2 | **50.9** | **32.3** | 40.6$^{+13.9}$ | 39.9 | 40.1 | **50.8** | **28.2** | 39.8$^{+16.2}$ |
| Qwen2.5-14B | 38.0 | 37.2 | 41.9 | 32.1 | 37.3 | 36.2 | 37.5 | 43.3 | 27.5 | 36.1 |
| + RLVR | 52.2 | 51.0 | **63.3** | 32.9 | 49.9$^{+12.6}$ | 53.2 | 52.1 | **64.2** | 30.5 | 50.0$^{+13.9}$ |
| + SPELL | **57.6** | **52.6** | 63.0 | **33.5** | **51.7**$^{+14.4}$ | **56.8** | **53.0** | 63.2 | **31.2** | **51.1**$^{+15.0}$ |
| Qwen2.5-32B | 46.8 | 42.6 | 49.0 | 33.7 | 43.0 | 40.7 | 42.2 | 50.1 | 28.7 | 40.4 |
| + RLVR | 58.3 | 50.0 | 59.5 | 32.8 | 50.2$^{+7.2}$ | 57.5 | 49.9 | 60.1 | 32.7 | 50.1$^{+9.7}$ |
| + SPELL | **61.8** | **50.2** | **62.1** | **34.2** | **52.1**$^{+9.1}$ | **60.6** | **52.2** | **62.3** | **34.3** | **52.4**$^{+12.0}$ |
| *Instruct Models* | | | | | | | | | | |
| Qwen2.5-7B-Instruct | 38.4 | 40.3 | 45.1 | 29.0 | 38.2 | 39.4 | 41.4 | 44.5 | 28.4 | 38.4 |
| + RLVR | 45.0 | **48.7** | 59.6 | 30.1 | 45.9$^{+7.7}$ | 44.1 | **48.6** | 57.4 | 28.2 | 44.6$^{+6.2}$ |
| + SPELL | **45.8** | 46.7 | **63.1** | **33.2** | **47.2**$^{+9.0}$ | **44.5** | 48.2 | **60.7** | **32.4** | **46.5**$^{+8.1}$ |
| Qwen2.5-14B-Instruct | 56.3 | 51.6 | 63.0 | 32.2 | 50.8 | 56.7 | 52.4 | 64.2 | 36.6 | 52.5 |
| + RLVR | 56.1 | 59.6 | 71.0 | 36.4 | 55.8$^{+5.0}$ | 56.6 | 59.9 | 73.4 | 38.5 | 57.1$^{+4.6}$ |
| + SPELL | **59.6** | **62.1** | **72.8** | **36.8** | **57.8**$^{+7.0}$ | **60.1** | **63.9** | **74.8** | **40.1** | **59.7**$^{+7.2}$ |
| Qwen2.5-32B-Instruct | 60.0 | 49.9 | 61.4 | 36.0 | 51.8 | 63.0 | 49.4 | 61.5 | 36.2 | 52.5 |
| + RLVR | 59.9 | 60.5 | 70.4 | 36.3 | 56.8$^{+5.0}$ | 59.7 | **62.3** | 69.6 | 36.9 | 57.1$^{+4.6}$ |
| + SPELL | **62.3** | **61.2** | **74.4** | **40.1** | **59.5**$^{+7.7}$ | **63.3** | 62.0 | **74.1** | **40.8** | **60.1**$^{+7.6}$ |
| Qwen3-30B-A3B-Instruct | 62.3 | 55.3 | 70.5 | 36.9 | 56.3 | 63.0 | 57.8 | 70.3 | 44.1 | 58.8 |
| + RLVR | 62.5 | 59.9 | 71.8 | 39.8 | 58.5$^{+2.2}$ | 64.0 | 62.0 | 72.4 | 47.4 | 61.5$^{+2.7}$ |
| + SPELL | **63.0** | **63.1** | **75.1** | **41.5** | **60.7**$^{+4.4}$ | **64.9** | **63.7** | **74.8** | **48.7** | **63.0**$^{+4.2}$ |
| Llama3.1-8B-Instruct | 33.2 | 45.6 | 52.5 | **29.1** | 40.1 | 34.9 | 47.3 | 53.5 | **27.1** | 40.7 |
| + RLVR | 37.9 | 45.0 | 58.8 | 27.5 | 42.3$^{+2.2}$ | 36.9 | 47.6 | 57.2 | 26.1 | 42.0$^{+1.3}$ |
| + SPELL | **39.2** | **48.9** | **61.6** | 28.4 | **44.5**$^{+4.4}$ | **39.7** | **50.8** | **60.9** | 26.2 | **44.4**$^{+3.7}$ |
| *Reasoning Models* | | | | | | | | | | |
| R1-Distill-Llama-8B | 42.0 | 50.3 | 66.8 | 27.9 | 46.8 | 41.5 | 52.6 | 69.3 | 26.4 | 47.5 |
| + RLVR | 43.4 | 51.4 | 67.8 | 30.0 | 48.2$^{+1.4}$ | 45.4 | 54.0 | 68.0 | 28.3 | 48.9$^{+1.4}$ |
| + SPELL | **48.9** | **53.4** | **68.4** | **30.2** | **50.2**$^{+3.4}$ | **49.2** | **54.3** | **70.0** | **29.3** | **50.7**$^{+3.2}$ |
| R1-Distill-Qwen-14B | 57.7 | 59.2 | 72.4 | 36.2 | 56.4 | 59.5 | 60.6 | 73.3 | 33.3 | 56.7 |
| + RLVR | 59.6 | 61.7 | 74.6 | 37.2 | 58.3$^{+1.9}$ | 61.0 | **63.8** | **76.0** | 35.9 | 59.2$^{+2.5}$ |
| + SPELL | **61.6** | **62.3** | **76.2** | **39.0** | **59.8**$^{+3.4}$ | **61.1** | 62.8 | 75.7 | **37.9** | **59.4**$^{+2.7}$ |
| Qwen3-4B-Thinking | 58.6 | **56.7** | 69.9 | 32.9 | 54.5 | 61.4 | 59.2 | 70.9 | 40.7 | 58.1 |
| + RLVR | 60.5 | 56.6 | 71.1 | 33.8 | 55.5$^{+1.0}$ | 63.3 | 58.6 | 71.1 | **43.4** | 59.1$^{+1.0}$ |
| + SPELL | **61.9** | 56.6 | **71.6** | **36.8** | **56.7**$^{+2.2}$ | **64.8** | **60.6** | **72.4** | 43.0 | **60.2**$^{+2.1}$ |
| Qwen3-30B-A3B-Thinking | 62.9 | 64.5 | 75.7 | 39.7 | 60.7 | 63.6 | 65.8 | 77.9 | 46.7 | 63.6 |
| + RLVR | 62.7 | 64.7 | 77.0 | 38.5 | 60.7$^{+0.0}$ | 63.9 | 67.1 | 77.2 | 49.6 | 64.5$^{+0.9}$ |
| + SPELL | **64.1** | **66.5** | **78.0** | **42.3** | **62.7**$^{+2.0}$ | **66.7** | **68.1** | **78.4** | **50.5** | **65.9**$^{+2.3}$ |

Figure 3: Test-time scaling performance (pass@k) across all benchmarks. The Qwen3-30B-A3B-Thinking model trained with SPELL shows a significantly steeper improvement as the number of samples (K) increases compared to the base model and the RLVR baseline. Notably, its pass@4 performance surpasses gemini-2.5-pro.

DocMath (-0.2) and LongBench-V2 (-1.2), whereas SPELL delivers consistent gains of 1.2 and 2.6 points, respectively. These results validate that when models approach or surpass the quality of static training data, a self-play curriculum proves more effective for sustaining performance gains.

**SPELL generalizes to longer contexts.** All models are trained with a 16K input limit and evaluated at 100K without additional tuning. The results remain consistent under this out-of-distribution input length, demonstrating that the benefits of SPELL extend beyond the training window. For Qwen2.5-14B, the average improvement is 14.4 at 16K and increases to 15.0 at 100K. This consistency suggests that the framework strengthens document-grounded reasoning in a way that remains effective as input lengths grow substantially, rather than producing gains limited to a specific context length.

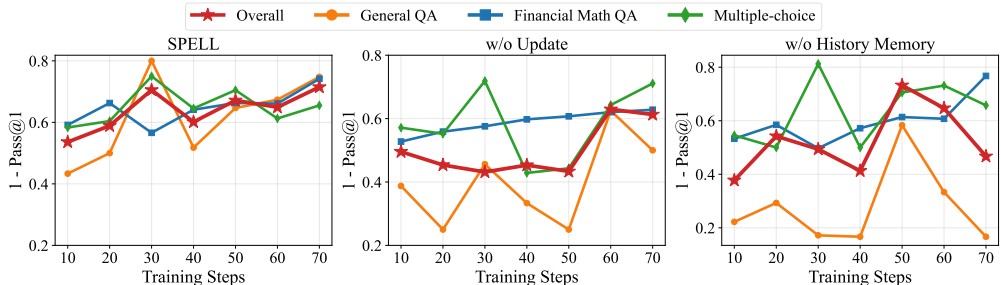

Figure 4: Analysis of question difficulty (1 - pass@1) on three tasks over training steps. **(Left)**: The full SPELL framework shows a clear upward trend in difficulty. **(Middle)**: Without questioner updates, difficulty stagnates. **(Right)**: Without the history memory, difficulty becomes erratic and unstable.

**SPELL boosts exploration and raises the performance ceiling.** We assess test-time exploration with the pass@k metric at a 100K input limit. As shown in Figure 3, Qwen3-30B-A3B-Thinking trained with SPELL exhibits a markedly steeper improvement curve as k increases compared to both the base model and the RLVR baseline. Its pass@8 score reaches 74.5, significantly outperforming the RLVR baseline (68.1) and the original base model (66.9). This enhanced exploratory ability further allows the SPELL-trained model to surpass the performance of the leading gemini-2.5-pro (Comanici et al., 2025) at a pass@4 rate. These results indicate that SPELL effectively broadens the model's test-time search space and raises its attainable performance ceiling, highlighting a promising path toward elevating the capabilities of even more powerful foundation models.

## 4.3 ABLATION STUDIES

To validate the key design choices within the SPELL framework, we conduct ablation studies on Qwen2.5-7B-Instruct. We individually remove each core component of the *questioner* and *verifier* roles to quantify their individual contributions to the overall performance.

**Questioner** As shown in Table 2, the removal of the format penalty and the grounding filter degrades the average score by 1.1 and 1.0 points, respectively. The format penalty keeps the question well-formed, and the grounding filter prevents the generation of hallucinated questions. The largest drops come from disabling the update mechanism and the history memory: freezing the questioner lowers the average score by 4.6, and removing history memory lowers it by 2.9. The declines appear across DocMath, Frames, LB-MQA, and LB-V2, which indicates that these components have a broad impact rather than task-specific effects.

Table 2: Ablation study of SPELL on Qwen2.5-7B-Instruct with a 16K maximum input length. - and + indicate relative decreases and increases, respectively, compared to the full SPELL model.

| Method | DocMath | Frames | LB-MQA | LB-V2 | Average |
|---|---|---|---|---|---|
| SPELL | 45.8 | 46.7 | 63.1 | 33.2 | 47.2 |
| *Questioner* | | | | | |
| *w/o* Format Penalty | $46.0^{+0.2}$ | $48.2^{+1.5}$ | $59.3^{-3.8}$ | $31.0^{-2.2}$ | $46.1^{-1.1}$ |
| *w/o* Grounding Filter | $47.0^{+1.2}$ | $46.4^{-0.3}$ | $60.1^{-3.0}$ | $31.3^{-1.9}$ | $46.2^{-1.0}$ |
| *w/o* Update | $45.5^{-0.3}$ | $43.8^{-2.9}$ | $50.9^{-12.2}$ | $30.3^{-2.9}$ | $42.6^{-4.6}$ |
| *w/o* History Memory | $45.6^{-0.2}$ | $46.6^{-0.1}$ | $54.2^{-8.9}$ | $30.8^{-2.4}$ | $44.3^{-2.9}$ |
| *Verifier* | | | | | |
| *w/o* Verifier | $39.4^{-6.4}$ | $46.6^{-0.1}$ | $60.4^{-2.7}$ | $29.4^{-3.8}$ | $44.0^{-3.2}$ |
| *w/o* Update | $45.1^{-0.7}$ | $48.2^{+1.5}$ | $61.4^{-1.7}$ | $32.6^{-0.6}$ | $46.8^{-0.4}$ |
| *w/o* Majority Voting | $45.5^{-0.3}$ | $48.1^{+1.4}$ | $61.9^{-1.2}$ | $30.7^{-2.5}$ | $46.6^{-0.6}$ |
| *w/o* Update Consistency | $46.6^{+0.8}$ | $47.1^{+0.4}$ | $57.7^{-5.4}$ | $31.3^{-1.9}$ | $45.7^{-1.5}$ |

indicates that these components have a broad impact rather than task-specific effects.

We further examine how these components affect generated question difficulty over Qwen2.5-7B-Instruct training steps, as measured by 1-pass@1 with an external responder (Qwen3-30B-A3B-Instruct) and an external verifier (gpt-oss-120b). The full SPELL model (Figure 4, left) shows a clear upward trend in overall question difficulty, which ensures the questioner proposes questions that are challenging enough for the responder's evolving capabilities. In contrast, freezing the questioner causes difficulty to stagnate (Figure 4, middle), while removing the history memory makes it erratic (Figure 4, right). The evidence supports the conclusion that continual updates and access to recent history are necessary to form a stable and progressively more challenging curriculum for the responder, which is essential for sustained improvement in a self-play system. This dynamic prevents one role from exploiting the static weaknesses of another, as observed in Liu et al. (2025a).

**Verifier** Removing the verifier and relying solely on rule-based rewards decreases average score by 3.2 points, with a 6.4-point drop on DocMath. The CEM-based reward function is brittle and can penalize semantically correct but lexically different answers; the verifier provides a complementary

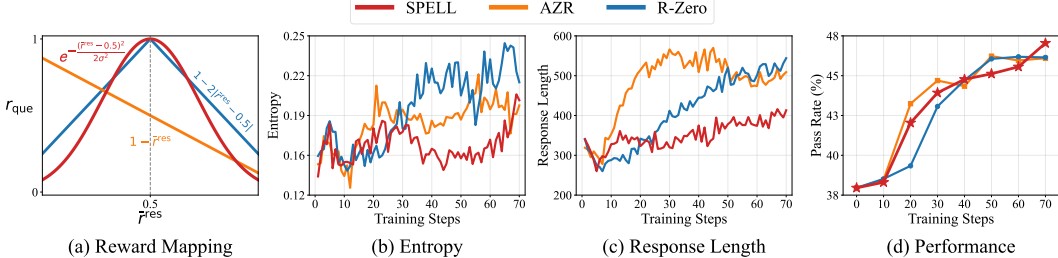

Figure 5: Comparison of different reward mapping strategies. **(a)** Visualization of the reward functions for SPELL, AZR, and R-Zero. **(b)** SPELL exhibits more stable entropy dynamics during training. **(c)** SPELL maintains a more moderate and controlled growth in response length. **(d)** These factors contribute to a consistent performance improvement, ultimately leading our method to achieve the highest final pass rate.

signal in such cases. Interestingly, disabling verifier updates or switching to single-pass decisions leads to moderate declines, which indicates that Qwen2.5-7B-Instruct is already competent at the simpler verification task. However, removing the consistency update mechanism still causes a 1.5-point performance drop. This result shows that the verifier's updates are susceptible to noise from its own erroneous majority votes, which degrades its reliability. On rule-verifiable tasks, the verifier learns to filter this noise by aligning its majority vote with the ground-truth rule-based outcome. This process provides the verifier with reliable learning signals, which in turn enhance its ability to generate stable rewards for rule-unverifiable outputs. This illustrates how verifiable rewards can guide the calibration of non-verifiable rewards, a finding that aligns with the self-judging methodology in Kimi-K2 (Team et al., 2025).

## 4.4 ANALYSIS OF QUESTIONER REWARD MAPPING

We compare our Gaussian-mapped reward function for questioner in Eq. (7) with the reward mapping used in AZR (Zhao et al., 2025) and R-Zero (Huang et al., 2025). Figure 5(a) visualizes these distinct mapping functions. While the AZR reward function also penalizes high-accuracy questions, it is susceptible to noise from spurious correctness, which can destabilize the training process. In contrast, our Gaussian function, which peaks when the average responder accuracy $\bar{r}^{res}$ is 0.5, selectively encourages questions at the frontier of the responder's competence. Additionally, this mechanism mitigates the impact of data noise. Questions with wrong reference answers typically result in success rates near zero or one, corresponding to scenarios of random guessing on unsolvable questions or consistent matching with the incorrect reference, respectively. Both extremes naturally fall into the low-reward tails of the Gaussian function, effectively suppressing incorrect questions during policy optimization. While R-Zero also centers its peak reward at 0.5, our Gaussian mapping provides a more targeted reward by offering stronger incentives for questions of moderate difficulty and imposing a steeper penalty on those that are either too easy or too hard. This creates a focused and smooth reward distribution that guides the questioner away from generating both trivial and overly difficult questions. The training dynamics corroborate these design differences. As shown in Figures 5(b) and (c), our method maintains a more stable training entropy and exhibits more controlled growth in response length under the Gaussian mapping than under AZR or R-Zero. These advantages in training stability lead to superior overall performance. As Figure 5(d) demonstrates, our method not only achieves more consistent performance growth but also reaches the highest final pass rate among all compared approaches. This evidence supports the view that concentrating the questioner's reward at the responder's competence frontier stabilizes the optimization process while preserving headroom for their mutual co-evolution.

## 4.5 HYPERPARAMETERS ANALYSIS

**Selection of standard deviation $\sigma$**  The choice of $\sigma$ in Eq. (7) is derived from the statistical properties of the Gaussian distribution, where approximately 99.7% of data points fall within three standard deviations ($3\sigma$) of the mean. In SPELL, we aim to concentrate the questioner's reward at the point of the responder's maximal learning efficiency, where $\bar{r}^{res} = 0.5$. Accordingly, the mean is set to $\mu = 0.5$. Given this mean, the distance to either boundary of the valid average responder reward range $[0, 1]$ is 0.5. By setting $3\sigma = 0.5$, we ensure the effective range of the questioner reward

covers the responder reward space, yielding $\sigma = 0.5/3$. To further validate this theoretical choice, we conduct an ablation study on $\sigma$ using Qwen2.5-7B-Instruct. As shown in Table 3, narrowing the curve ($\sigma = 0.5/6$) has a minimal negative impact, as the reward remains well-focused. However, widening the curve ($\sigma = 0.5/2$) significantly degrades performance, likely because it assigns higher rewards to overly easy or hard questions, providing a less targeted training signal. This confirms that $\sigma = 0.5/3$ is both theoretically sound and empirically effective.

Table 3: Ablation analysis of SPELL varying the standard deviation $\sigma$ and the rollout group size $G$ using Qwen2.5-7B-Instruct. The default configuration is $\sigma = 0.5/3$ and $G = 8$.

| Settings | DocMath | Frames | LB-MQA | LB-V2 | Average |
|---|---|---|---|---|---|
| SPELL | 45.8 | 46.7 | **63.1** | **33.2** | **47.2** |
| *Standard Deviation* ($\sigma$) | | | | | |
| $\sigma = 0.5/6$ | 45.3 | 47.0 | 62.3 | 32.8 | 46.9 |
| $\sigma = 0.5/2$ | 45.5 | 46.0 | 59.4 | 31.8 | 45.7 |
| *Group Size* ($G$) | | | | | |
| $G = 4$ | 44.3 | 47.1 | 62.5 | 31.8 | 46.4 |
| $G = 16$ | **46.0** | **47.4** | 62.7 | 31.9 | 47.0 |

**Sensitivity of group size** ($G$)   We examine the impact of the rollout group size $G$ on model performance using Qwen2.5-7B-Instruct. As shown in Table 3, while $G = 8$ yields the best overall results, SPELL remains robust across different group sizes. We select $G = 8$ as the default setting to strike a balance between performance gains and computational efficiency during training.

### 4.6   ROLE OF EXTERNAL JUDGES IN VERIFICATION

We investigate whether replacing the rule-based judge (CEM-based reward function) with a stronger external model (gpt-oss-120b) benefits the self-play process. As shown in Table 4, introducing a stronger external judge does not yield a significant overall improvement. This suggests that Qwen2.5-7B-Instruct is already capable of learning semantic verification through self-play without external supervision. Notably, when an external judge is introduced, the internal verifier becomes less important; removing it results in only a minor 0.5-point drop, compared to the significant 3.2-point drop observed when using the rule-based judge. This highlights the critical role of the internal verifier in complementing the brittle CEM-based reward function when an external judge is not available.

Table 4: Comparison of SPELL trained with rule-based judge versus an external judge (gpt-oss-120b). The verifier is crucial when using a rule-based judge, but becomes less critical when including an external judge.

| Method | DocMath | Frames | LB-MQA | LB-V2 | Average |
|---|---|---|---|---|---|
| Qwen2.5-7B-Instruct | 38.4 | 40.3 | 45.1 | 29.0 | 38.2 |
| + SPELL (Rule-based Judge) | 45.8 | 46.7 | **63.1** | **33.2** | **47.2** |
| + SPELL (Gpt-oss-120b Judge) | **47.1** | **48.0** | 61.6 | 32.1 | **47.2** |
| + SPELL (Rule-based Judge) *w/o* Verifier | 39.4 | 46.6 | 60.4 | 29.4 | 44.0 |
| + SPELL (Gpt-oss-120b Judge) *w/o* Verifier | 47.0 | 47.2 | 61.1 | 31.3 | 46.7 |

## 5   CONCLUSION

This work introduces SPELL, a multi-role self-play reinforcement learning framework for evolving the long-context reasoning capabilities of LLMs without human supervision. A single policy model alternates among the roles of questioner, responder, and verifier to generate questions, solve them, and assess the solutions, which reduces reliance on costly and unreliable human annotation while enabling stable self-evolution. Extensive experiments across 12 models of diverse architectures and sizes show that SPELL delivers consistent and substantial improvements in long-context reasoning.

This study concludes with three notable findings. First, signals from verifiable tasks can calibrate and strengthen the verifier's assessment on non-verifiable tasks, thereby ensuring a reliable self-rewarding mechanism. Second, within a multi-role self-play framework, sustaining a dynamic equilibrium among the capabilities of different roles is critical for the stable evolution of the shared policy. Finally, our results demonstrate that for models approaching or surpassing human performance, where external supervision emerges as a fundamental bottleneck, autonomous self-evolution transitions from a promising alternative to an indispensable strategy for sustained advancement.

ETHICS STATEMENT

This research focuses on the development of long-context LLMs through self-play that requires no human supervision. While we believe our methodology does not inherently raise significant ethical issues, we acknowledge the potential for misuse of this technology. We also recognize that an unsupervised learning approach may perpetuate or amplify societal biases in the model. Our research is conducted using only publicly available datasets, in compliance with their licenses, and involves no personally identifiable information. We have adhered to all relevant ethical and legal standards and declare no conflicts of interest that could have influenced the outcomes of this study.

REPRODUCIBILITY STATEMENT

To ensure the reproducibility of our work, we provide full experimental details in Section 4.1 and Appendix E. These include our methods for dataset construction, training configurations, and the evaluation setup. The implemented code, the data used, and a comprehensive guide to reproduce our method are available in the supplementary materials.

ACKNOWLEDGEMENTS

This work was supported by Alibaba Group through Alibaba Research Intern Program and National Natural Science Foundation of China (No. 62576368).

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

## A    STATEMENT ON THE USE OF LARGE LANGUAGE MODELS

In preparing this manuscript, we employ large language models (LLMs) purely as an assistive writing tool, without influencing the intellectual contributions of this work. Their use is limited to checking grammar, correcting spelling, and improving the clarity and precision of the text. The proposal of the research question, the development of the methodology, and the experimental design are the original contributions of the authors. All model-generated outputs are subject to critical review, editing, and final verification by the authors to ensure the authenticity of the content.

## B    LIMITATIONS AND FUTURE WORK

First, while our study provides strong empirical evidence, a theoretical framework explaining the co-evolutionary dynamics of the three roles within a single model has yet to be explored. Second, due to framework and efficiency constraints, our experiments are limited to a 16K input context. Although the acquired skills generalize well to longer contexts, it is a critical next step to develop more efficient frameworks for self-play RL on ultra-long contexts, such as 128K tokens and beyond. Finally, our self-play framework still relies on a degree of human intervention, such as pre-processing the document corpus and crafting prompt templates for each role. Future work could explore pathways toward greater autonomy, such as a system where an LLM interacts with a real-world environment to generate and evolve its own tasks, templates, and reward functions.

## C    RELATED WORK

### C.1    LONG-CONTEXT ALIGNMENT

Developing long-context language models (LCLMs) has become a central research area, as many real-world applications require reasoning over extended inputs (Liu et al., 2025b). A dominant paradigm in this field is to enhance models through various post-training alignment techniques on well-curated, synthesized datasets. One prominent approach is supervised fine-tuning (SFT). For instance, LongAlign (Bai et al., 2024a) utilizes a self-instruct pipeline to construct a large-scale, long-instruction dataset for SFT, while MIMG (Chen et al., 2025c) employs a multi-agent system to generate more complex, multi-hop reasoning data. Another line of work focuses on preference optimization. LongPO (Chen et al., 2025a) and SoLoPO (Sun et al., 2025) generate preference pairs by contrasting outputs from compressed versus full contexts and leverage direct preference optimization (DPO) (Rafailov et al., 2023) to transfer short-context capabilities to longer inputs. More recently, QwenLong-L1 (Wan et al., 2025) introduces the concept of long-context reasoning RL and employs a two-stage progressive context scaling training to develop long-context reasoning models. While proven effective, all these approaches exhibit some form of reliance on external supervision.

### C.2    SELF-PLAY LANGUAGE MODELS

To mitigate the reliance on external supervision, such as human annotation or labeled datasets, researchers are developing self-play language models (Gao et al., 2025). These models achieve autonomous improvement by generating their own training data, reward signals, or both. These approaches can be categorized into two paradigms: multi-model optimization, which co-evolves several distinct models, and single-model optimization, where one model assumes multiple roles.

**Multi-Model Optimization**    This paradigm orchestrates the co-evolution of multiple specialized models. In Mutual-Taught (Shi et al., 2025), a policy model and a reward model are reciprocally and iteratively refined: the policy model generates data to enhance the reward model, which in turn provides more accurate feedback to improve the policy model. Similarly, in R-Zero (Huang et al., 2025), a challenger and a solver LLM are independently optimized and co-evolve: the challenger generates new, challenging math problems for which it is rewarded based on the solver's uncertainty, and the solver is fine-tuned on these questions, rewarded for correctly solving them. While effective, this paradigm substantially increases systemic complexity, and performance gains often plateau after a finite number of iterations.

**Single-Model Optimization**    In contrast to the multi-model approach, single-model optimization reduces systemic complexity by leveraging a single model to assume multiple roles for self-improvement. For instance, Absolute Zero Reasoner (AZR) (Zhao et al., 2025) employs one model as

both a proposer that generates complementary coding tasks (induction, abduction, and deduction) and a solver that addresses them, with code execution feedback serving as the reward signal. Similarly, the Self-Challenging Agent (SCA) (Zhou et al., 2025) employs a model to formulate novel "Code-as-Task" questions and subsequently solve them, using self-generated code functions to provide the verification signal. A significant limitation of such approaches, however, is their dependence on external code executors, which confines their applicability to domains with programmatically verifiable outcomes. To overcome this limitation and enhance autonomy, other approaches further incorporate self-rewarding mechanisms that leverage self-consistency (Zuo et al., 2025), internal confidence scores (Li et al., 2025), or self-generated evaluations (Yuan et al., 2024). For example, Self-Questioning Language Models (SQLM) (Chen et al., 2025b) utilize a model to propose and then answer questions, adapting its reward mechanism between self-consistency for arithmetic and proposer-generated unit tests for coding. Similarly, the Self-Rewarding Self-Improving framework (Simonds et al., 2025) also generates its own questions and solutions but uses a self-judging mechanism for reward computation.

Our work extends the single-policy self-play paradigm to long-context understanding and reasoning. Unlike existing methods that focus on short-context tasks like coding or math, SPELL is designed for reasoning over long documents. In our framework, a single LLM learns by playing three roles: a *questioner*, a *responder*, and a *verifier*. These roles interact to create a self-sufficient learning loop for long-context comprehension, thereby addressing a key gap in current self-play learning research.

## D  SPELL ALGORITHM

In this section, we outline the step-by-step algorithm for SPELL in Algorithm 1.

---

**Algorithm 1** The SPELL Algorithm

---

1: **Require:** Initial policy model $\pi_\theta$; Dataset $\mathcal{D} = \{(C, \tau)\}$; Subset size $m$; History memory size $L$; Batch size $N$; Group size $G$
2: **for** $(C, \tau) \in \mathcal{D}$ **do**
3:     $\mathcal{H}_C \leftarrow \text{Queue}(\emptyset, L)$                   ▷ Initialize a history memory for each document cluster
4: **for** each training step $t = 1, 2, \ldots$ **do**
5:     $\mathcal{B}^{\text{que}}, \mathcal{B}^{\text{res}}, \mathcal{B}^{\text{ver}} \leftarrow \emptyset, \emptyset, \emptyset$             ▷ Initialize empty sample batches for three roles
6:     **while** $|\mathcal{B}^{\text{res}}| < N$ **do**
7:         $(C, \tau) \sim \mathcal{D}$                   ▷ Sample a document cluster and task type
8:         $C_{\text{new}} \leftarrow \text{SampleDocs}(C, m)$       ▷ Sample a subset of m documents for questioner
9:         $X^{\text{que}} \leftarrow \text{GetQuestionerContext}(C_{\text{new}}, \tau, \mathcal{H}_C)$    ▷ Prepare questioner input; see §3.1
10:        $(q, a) \sim \pi_\theta^{\text{que}}(\cdot | X^{\text{que}})$     ▷ **Questioning**: Generate a question with reference answer
11:        $y_{\text{no\_context}} \sim \pi_\theta^{\text{res}}(\cdot | q)$    ▷ Attempt to answer the question without document context
12:        **if** $\mathcal{R}_{\text{rule}}(y_{\text{no\_context}}, a) = 1$ **then**                  ▷ Grounding filter
13:           $\mathcal{B}^{\text{que}} \leftarrow \mathcal{B}^{\text{que}} \cup \{(X^{\text{que}}, q, a, r^{\text{que}} = -0.5)\}$ ▷ Penalize and store ungrounded question
14:          **continue**                  ▷ Discard question and proceed to the next iteration
15:        $\{y_i\}_{i=1}^G \sim \pi_\theta^{\text{res}}(\cdot | C, q)$          ▷ **Responding**: Generate a group of $G$ responses
16:        **for** $i \leftarrow 1$ to $G$ **do**
17:          $\{v_{i,j}^{\text{ver}}\}_{j=1}^G \sim \pi_\theta^{\text{ver}}(\cdot | q, y_i, a)$ ▷ **Verifying**: Generate $G$ verifications for each response
18:        $r^{\text{que}}, \{r_i^{\text{res}}\}_i, \{\{r_{i,j}^{\text{ver}}\}\}_{i,j} = \text{ComputeReward}(q, a, \{y_i\}_i^{\text{res}}, \{\{v_{i,j}^{\text{ver}}\}\}_{i,j})$
19:                               ▷ Compute role-specific rewards; see Eq. (4)-Eq. (7)
20:        **if** $r^{\text{que}} > 0$ **then**        ▷ Append solvable questions to the history memory
21:          $\text{PushToHistoryMemory}(\mathcal{H}_C, (C_{\text{new}}, q, a))$
22:        $\mathcal{B}^{\text{que}} \leftarrow \mathcal{B}^{\text{que}} \cup \{(X^{\text{que}}, q, a, r^{\text{que}})\}$
23:        $\mathcal{B}^{\text{res}} \leftarrow \mathcal{B}^{\text{res}} \cup \{(C, q, \{y_i\}_i, \{r_i^{\text{res}}\}_i)\}$
24:        $\mathcal{B}^{\text{ver}} \leftarrow \mathcal{B}^{\text{ver}} \cup \{(q, a, \{y_i\}_i, \{v_{i,j}\}_{i,j}, \{r_{i,j}^{\text{ver}}\}_{i,j})\}$
25:     $\mathcal{B} \leftarrow \text{SampleBatch}(\mathcal{B}^{\text{que}}, \mathcal{B}^{\text{res}}, \mathcal{B}^{\text{ver}})$     ▷ Apply role-specific dynamic sampling; see §3.3
26:     $\pi_\theta \leftarrow \text{UpdatePolicy}(\pi_\theta, \mathcal{B})$                  ▷ **Unified Policy Update**; see §3.3
27: **Ensure:** Updated policy model $\pi_\theta$

---

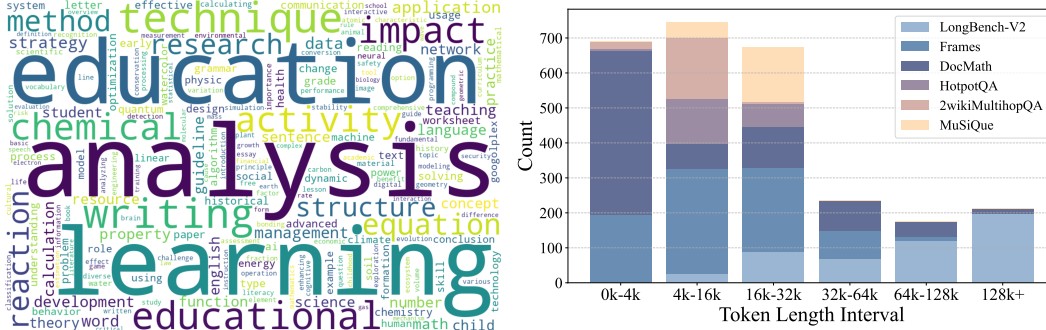

Figure 6: **(Left)** The word cloud of training document labels, with font size indicating frequency. The prominence of terms like *education*, *analysis*, and *learning* underscores the dataset's focus on knowledge-intensive content. **(Right)** The token length distribution, calculated by the Qwen2 tokenizer, shows that our evaluation benchmarks cover a wide spectrum of context lengths.

# E    IMPLEMENTATION DETAILS

## E.1    TRAINING DATA CONSTRUCTION

Our training data supports three distinct tasks: document general QA, financial math QA, and multiple-choice QA. We construct the dataset from two complementary sources. The first is the DocMath dataset (Zhao et al., 2024), which provides specialized data comprising long, complex financial reports that necessitate numerical reasoning. We used only the raw documents, discarding the original unlabeled questions. From this dataset, we select 2,150 instances with a total token length below 16K, which are designated for the financial math QA task.

The second component is a general-domain corpus designed to enhance the model's fundamental document understanding. This dataset originates from the 1.16 billion English documents in the Ultra-Fineweb corpus (Wang et al., 2025) and is curated through the following multi-stage pipeline. First, an initial filtering stage retains high-quality texts of appropriate length by selecting documents with a perfect quality score of 1.0 and character lengths between 100 and 32,768. This step downsamples the corpus to a high-quality subset of 4 million documents. Second, a cleaning procedure ensures data diversity and prevents test set contamination. We employ the MinHashLSH algorithm (Shrivastava & Li, 2014) with a threshold of 0.8 and 128 permutation functions for deduplication and to decontaminate the data against all documents in our evaluation benchmarks. This cleaning phase refines the corpus to 1.1 million unique and decontaminated texts. Third, a hierarchical clustering approach structures the dataset into different topics. We generate 4096-dimensional embeddings for each text using the Qwen3-Embedding-8B model (Zhang et al., 2025), which are then grouped into 50,000 distinct clusters via a hierarchical k-means algorithm (Xu et al., 2015). Each cluster contains an average of 20 semantically related documents, and the resulting domain distribution, visualized in Figure 6 (left), confirms the dataset's broad topical diversity. This curated corpus from Ultra-Fineweb is used to generate training instances for the document general QA and multiple-choice tasks. To ensure a comparable scale across tasks, the number of clusters selected for these two tasks matches the data size of the DocMath portion. Ultimately, our combined training data consists of 6,450 unique document sets, with 2,150 designated for each of the three tasks.

## E.2    EVALUATION DETAILS

**Evaluation Benchmarks**    We evaluate our models using a suite of well-established benchmarks designed to assess long-context comprehension and reasoning. These benchmarks fall into two primary categories: multiple-choice and multi-hop question answering (QA). For the multiple-choice task, we use LongBench-V2 (Bai et al., 2025), a benchmark of 503 questions that assesses deep comprehension across six areas: single-document QA, multi-document QA, long in-context learning, long-dialogue history understanding, code repository understanding, and long structured data understanding. For multi-hop QA, our evaluation incorporates several benchmarks: Frames (Krishna et al., 2025), containing 824 questions on diverse Wikipedia topics such as history, sports, science, animals, and health; three subsets from LongBench (Bai et al., 2024b), each with 200 questions, namely HotpotQA (Yang et al., 2018) (2-hop), 2WikiMultihopQA (Ho et al., 2020) (requiring up

to five hops), and MuSiQue (Trivedi et al., 2022) (requiring up to four hops); and DocMath (Zhao et al., 2024), which focuses on numerical reasoning within financial reports. For DocMath, we use the testmini subset of 800 queries, which is orthogonal to our training data. As shown in Figure 6 (right), the test instances across these benchmarks cover a wide range of context lengths.

**Evaluation Configurations** We evaluate all models at two maximum input lengths: 16K tokens, which aligns with our training configuration, and 100K tokens to test for generalization to longer contexts. The maximum generation length is 4K tokens for non-reasoning models and is extended to 20K tokens for reasoning models. For prompts exceeding the maximum context window, we employ the middle truncation strategy from Bai et al. (2024b) to preserve the front and tail portions of the context. All experiments are conducted using a sampling temperature of 0.7 and a top-$p$ value of 0.95. For each query, we generate $n = 8$ candidate responses, reporting the average score (pass@1) for our main experiments and the pass@k metric for the test-time scaling analysis. The pass@k metric is an unbiased estimator for the probability that at least one of $k$ candidate solutions is correct, given $n$ candidates per problem, of which $c$ are correct. It is calculated as:

$$\text{pass@k} = 1 - \frac{\binom{n-c}{k}}{\binom{n}{k}} \tag{11}$$

Our scoring is tailored to each benchmark's format. For multiple-choice tasks, we report standard accuracy. For multi-hop QA tasks, simple string matching is often insufficient to assess the semantic correctness of free-form text answers. Thus, we supplement the cover exact match (CEM) score with LLM-as-a-judge (Zheng et al., 2023), which uses gpt-oss-120b (OpenAI, 2025) to evaluate semantic equivalence between a model's prediction and the ground-truth answer. The prompt for this evaluation is detailed in Table 5. The final score for these tasks is the maximum of the two, providing a more comprehensive and robust assessment of model performance.

Table 5: Prompt template for LLM-as-a-judge to compare the predicted answer and the ground truth given the question. Modified from Frames (Krishna et al., 2025).

---

**LLM Judge Prompt for Multi-Hop QA Tasks.**

## TASK
I need your help in evaluating an answer provided by an LLM against a ground truth answer. Your task is to determine if the ground truth answer is present in the LLM's response. Please analyze the provided data and make a decision.
## Instruction
1. Carefully compare the "Predicted Answer" with the "Ground Truth Answer".
2. Consider the substance of the answers - look for equivalent information or correct answers. Do not focus on exact wording unless the exact wording is crucial to the meaning.
3. Your final decision should be based on whether the meaning and the vital facts of the "Ground Truth Answer" are present in the "Predicted Answer".
4. Your decision **must be** one of the "[[YES]]" or "[[NO]]".
## Input Data
- Question: {question}
- Predicted Answer: {predicted answer}
- Ground Truth Answer: {ground truth}
## Output Format
Provide your final evaluation in the following format:
"Explanation:" "How you made the decision"
"Decision:" "[[YES]]" or "[[NO]]"
Please proceed with the evaluation.

---

### E.3 RL ALGORITHM DETAILS

To enhance stability and practical performance, we integrate two key techniques inspired by Decoupled Clip and Dynamic Sampling Policy Optimization (DAPO) (Yu et al., 2025) for the baselines and our method. First, we adopt a **token-level policy gradient loss**, which normalizes each token's contribution by the total number of tokens in the group. This approach ensures every token in the

same group contributes equally to the final objective, which prevents the learning signal from valuable tokens in high-quality, long responses from being diluted while ensuring that undesirable patterns in low-quality, lengthy outputs are effectively penalized. Second, we employ a **dynamic sampling** strategy: any group of generations where the rewards $\{r_i\}_{i=1}^{G}$ exhibit zero variance is discarded from the training batch. This ensures the advantage in Eq. (3) is always well-defined.

Consistent with recent findings suggesting that removing KL regularization can improve exploration and accelerate convergence (Hu et al., 2025; Yu et al., 2025; Wan et al., 2025), we set $\beta = 0$. Besides, we operate in a strictly on-policy setting, performing only a single gradient update per batch of samples. This design choice implies that the policy being updated, $\pi_\theta$, remains identical to the policy that generated the data, $\pi_{\theta_{\text{old}}}$. Since the importance sampling ratio $\rho_{i,t}(\theta)$ is strictly equal to 1, the clipping function becomes inactive, and we can remove it from the objective. Note that the advantage $A_i$ is independent of $t$, the training objective in Eq. (2) simplifies to:

$$
\mathcal{J}_{\text{GRPO}}(\theta) = \mathbb{E}_{c,q\sim\mathcal{D},\{y_i\}_{i=1}^{G}\sim\pi_{\theta_{\text{old}}}} \left[ \frac{1}{\sum_{j=1}^{G}|y_j|} \sum_{i=1}^{G} A_i \sum_{t=1}^{|y_i|} \rho_{i,t}(\theta) \right].
\tag{12}
$$

### E.4 Details of Open-Source Models and the Dataset

In Table 6, we provide the Huggingface repository names of all policy models, the embedding model, the judge model, and datasets used in our experiments.

Table 6: Details of open-source models and datasets in our experiments.

| Model/Dataset | Huggingface ID |
|---|---|
| Qwen2.5-7B | Qwen/Qwen2.5-7B |
| Qwen2.5-14B | Qwen/Qwen2.5-14B |
| Qwen2.5-32B | Qwen/Qwen2.5-32B |
| Qwen2.5-7B-Instruct | Qwen/Qwen2.5-7B-Instruct |
| Qwen2.5-14B-Instruct | Qwen/Qwen2.5-14B-Instruct |
| Qwen2.5-32B-Instruct | Qwen/Qwen2.5-32B-Instruct |
| Qwen3-30B-A3B-Instruct | Qwen/Qwen3-30B-A3B-Instruct-2507 |
| Llama-3.1-8B-Instruct | meta-llama/Meta-Llama-3.1-8B-Instruct |
| R1-Distill-Llama-8B | deepseek-ai/DeepSeek-R1-Distill-Llama-8B |
| R1-Distill-Qwen-14B | deepseek-ai/DeepSeek-R1-Distill-Qwen-14B |
| Qwen3-4B-Thinking | Qwen/Qwen3-4B-Thinking-2507 |
| Qwen3-30B-A3B-Thinking | Qwen/Qwen3-30B-A3B-Thinking-2507 |
| Qwen3-Embedding-8B | Qwen/Qwen3-Embedding-8B |
| gpt-oss-120b | openai/gpt-oss-120b |
| DocMath | yale-nlp/DocMath-Eval |
| Ultra-Fineweb | openbmb/Ultra-FineWeb |

### E.5 Details of Baselines

We compare SPELL with two categories of baselines. The first category comprises the base models on which SPELL is built, organized into three groups: (1) **base models**, including Qwen-2.5-7B, Qwen-2.5-14B, and Qwen-2.5-32B; (2) **instruction-tuned models**, including Qwen-2.5-7B-Instruct, Qwen-2.5-14B-Instruct, Qwen-2.5-32B-Instruct, Qwen3-30B-A3B-Instruct, and Llama-3.1-8B-Instruct; and (3) **reasoning models**, including R1-Distill-Llama-8B, R1-Distill-Qwen-14B, Qwen3-4B-Thinking, and Qwen3-30B-A3B-Thinking (Yang et al., 2025; 2024a; Guo et al., 2025; Dubey et al., 2024). The second baseline is traditional RLVR. For data construction, we use DeepSeek-R1-0528 (Guo et al., 2025) as both the questioner and responder to synthesize a dataset from the same document clusters used for SPELL. The synthesized dataset is then filtered by a verifier model (gpt-oss-120b), and retains only the instances where the answers from the questioner and responder are identical. This process yields a dataset of approximately 3,000 verifiable samples that covers multiple-choice, document general QA, and financial math QA tasks. For RLVR training, we employ the cover exact match (CEM) as the rule-based reward function. We generate eight trajectories per question and maintain all other hyperparameters identical to those of SPELL to ensure a fair comparison.

## F  ADDITIONAL ANALYSIS

This section provides further insights into the properties of SPELL. We begin by analyzing the training cost in Section F.1 and exploring generalization to short-context tasks in Section F.2. We then extend our evaluation to additional long-context benchmarks in Section F.3 and compare our approach with state-of-the-art long-context alignment baselines in Section F.4. The section concludes with an analysis of the training dynamics of role evolution in Section F.5 and a discussion on reward hacking risks and mitigations in Section F.6.

### F.1  TRAINING COST ANALYSIS

We analyze the computational cost of SPELL using the Qwen2.5-7B-Instruct model on a single node with $8 \times 80$GB NVIDIA A100 GPUs. Figure 7 illustrates the time breakdown for the two primary stages of our framework: role-specific rollout (questioning, responding, and verifying) and unified updating. The total time per training step averages approximately seven minutes, and the total training cost is about eight hours. Although the number of verifier rollouts is $G$ times greater than that of the responder, as outlined in Section 3.3, its computational cost is lower. This efficiency stems from two factors: the verifier processes shorter inputs, as it does not require the long documents provided to the responder, and its task of generating brief verifications is less demand-

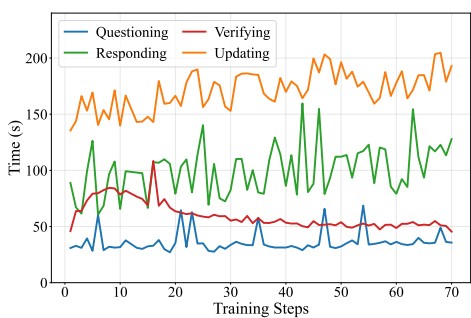

Figure 7: Time breakdown per training step for the four main stages of SPELL. Verifying constitutes a small amount of the total cost.

ing than the responder's task of reasoning over long contexts. During the later stages of training, the time for verifying constitutes only about half of the time required for responding. Consequently, generating $G$ verification judgments for each response to create a reliable reward signal does not introduce a significant computational bottleneck. As shown in Section 4.3 and Table 2, the verifier provides a significant 3.2-point performance gain (from 44.0 to 47.2). The significant performance gain by introducing the verifier far outweighs the minor increase in training cost.

### F.2  SHORT-CONTEXT REASONING RESULTS

We further investigate the generalization of our proposed SPELL method from long-context reasoning to short-context reasoning tasks. Our evaluation suite includes five mathematical benchmarks and the MMLU Pro general knowledge benchmark as follows:

**AIME 24/25** (MAA, 2025) is the American Invitational Mathematics Examination, a prestigious high-school mathematics competition administered by the Mathematical Association of America (MAA). The AIME consists of two exams (I and II) annually, each containing 15 problems, for a total of 30 problems per year. The answer to each question is an integer from 0 to 999. These problems demand deep mathematical knowledge and creative problem-solving strategies, making them a challenging benchmark for advanced mathematical reasoning.

**AMC 23**[4] (Yang et al., 2024b) refers to the 2023 American Mathematics Competition. This competition features 25 questions designed to test advanced high school mathematics, covering topics such as trigonometry, advanced algebra, and advanced geometry.

**MATH** (Hendrycks et al., 2021b) is a dataset of math problems ranging in difficulty from middle school to high school competition level. It is designed to test a wide range of mathematical skills, including algebra, geometry, number theory, and counting & probability. For our evaluation, we utilize a subset of 500 problems, referred to as MATH-500.

**GSM8K** (Cobbe et al., 2021) is a dataset of 1,319 grade school math word problems. These problems are designed to be solvable by a capable middle school student and require two to eight steps of reasoning using basic arithmetic operations.

---

[4]https://huggingface.co/datasets/AI-MO/aimo-validation-amc

**MMLU-Pro** (Wang et al., 2024) is an enhanced version of the MMLU (Hendrycks et al., 2021a) dataset, designed to address issues such as noisy data and reduced difficulty resulting from advances in model capabilities and increased data contamination. MMLU-Pro raises the difficulty by expanding the multiple-choice options from four to as many as ten and incorporating expert-reviewed annotations for improved quality and reduced noise.

Table 7: Evaluation results for base models on short-context reasoning tasks.

| Model | AIME24 | AIME25 | AMC23 | MATH500 | GSM8K | MMLU Pro | Average |
|---|---|---|---|---|---|---|---|
| Qwen2.5-7B | 5.42 | 3.33 | 33.44 | 53.60 | 76.57 | 40.24 | 35.43 |
| + SPELL | $9.17^{+3.75}$ | $5.00^{+1.67}$ | $40.31^{+6.8}$ | $63.60^{+10.00}$ | $86.28^{+9.71}$ | $49.78^{+9.54}$ | $42.36^{+6.93}$ |
| Qwen2.5-14B | 6.67 | 5.83 | 37.81 | 61.68 | 84.31 | 46.67 | 40.50 |
| + SPELL | $12.08^{+5.41}$ | $10.42^{+4.59}$ | $50.31^{+12.}$ | $72.40^{+10.72}$ | $91.36^{+7.05}$ | $58.86^{+12.19}$ | $49.24^{+8.74}$ |
| Qwen2.5-32B | 9.17 | 5.83 | 45.31 | 66.25 | 87.34 | 48.89 | 43.80 |
| + SPELL | $15.83^{+6.66}$ | $8.33^{+2.50}$ | $55.62^{+10.}$ | $76.00^{+9.75}$ | $90.25^{+2.91}$ | $60.22^{+11.33}$ | $51.04^{+7.24}$ |

Consistent with our main experiments, all models are evaluated with a maximum output of 4K tokens, a sampling temperature of 0.7, and a top-$p$ value of 0.95. We report the accuracy averaged over 8 independent runs for each task. As shown in Table 7, SPELL, trained solely on the long-context data, improves performance on short-context reasoning benchmarks across all base models. The average scores of Qwen2.5-7B, Qwen2.5-14B, and Qwen2.5-32B increase by 6.93, 8.74, and 7.24 points, respectively. The consistent gains indicate that reasoning competencies acquired through long-context self-play transfer effectively to short-context settings.

### F.3 ADDITIONAL LONG-CONTEXT BENCHMARKS

We further evaluate SPELL on two challenging long-context benchmarks: MRCR (Vodrahalli et al., 2024)[5], a multi-needle "Needle in a Haystack" benchmark, and three subsets of HELMET (Yen et al., 2025), which covers Retrieval-Augmented Generation (RAG), In-Context Learning (ICL), and Summarization (Summ). We evaluate the Qwen2.5 base models against the RLVR baseline and SPELL with a maximum input length of 16K and maximum output length of 4K. The results in Table 8 show that SPELL consistently and significantly outperforms both the base models and the RLVR baseline across these diverse long-context tasks, demonstrating strong generalization capabilities beyond the standard QA tasks.

Table 8: Evaluation results for base models on MRCR and HELMET subsets. The best score in each model group is highlighted in **bold**.

| Model | MRCR-2needle | MRCR-4needle | MRCR-8needle | Helmet-RAG | Helmet-ICL | Helmet-Summ | Average |
|---|---|---|---|---|---|---|---|
| Qwen2.5-7B | 6.9 | 2.0 | 2.2 | 50.0 | 3.5 | 4.1 | 11.5 |
| + RLVR | 22.0 | 12.5 | 10.5 | 49.3 | 2.6 | **14.3** | 18.5 |
| + SPELL | **34.5** | **16.5** | **16.0** | **54.2** | 10.4 | 13.7 | **24.2** |
| Qwen2.5-14B | 23.1 | 10.5 | 10.0 | 42.4 | 1.5 | 3.7 | 15.2 |
| + RLVR | 20.9 | 9.5 | 9.1 | 46.7 | **42.0** | **24.7** | 25.5 |
| + SPELL | **35.0** | **22.1** | **17.8** | **52.3** | 39.2 | 23.0 | **31.6** |
| Qwen2.5-32B | 27.0 | 11.5 | 12.0 | 59.0 | 42.8 | 16.2 | 28.1 |
| + RLVR | 36.7 | 14.6 | 13.0 | 52.7 | 16.7 | 21.0 | 25.8 |
| + SPELL | **38.0** | **18.5** | 14.7 | **61.4** | **56.4** | **21.2** | **35.0** |

### F.4 COMPARISON WITH LONG-CONTEXT ALIGNMENT BASELINES

We compare SPELL against three recent long-context alignment baselines—SoLoPO (Sun et al., 2025), LongPO (Chen et al., 2025a), and QwenLong-L1 (Wan et al., 2025)—using Qwen2.5-7B-Instruct as the base model. To ensure a fair comparison, we reimplement these methods using the same document corpus employed in SPELL. For SoLoPO and LongPO, the core step is to construct preference pairs from short contexts containing key information and long contexts containing distractors given the same question. Specifically, for each data instance comprising $n$ documents, we first randomly sample $m = 5$ documents as the short text and employ DeepSeek-R1-0528 as

---
[5]https://huggingface.co/datasets/openai/mrcr

the questioner to generate QA pairs. Then, we use DeepSeek-R1-0528 as the responder to answer the proposed questions using the full set of $n$ documents. We retain only those QA pairs where the answers from the questioner and responder are consistent. Next, we take the 5 documents from the questioning stage as short texts, corresponding to $x_{\text{short}}$ in SoLoPO and $x_S$ in LongPO, and take all $n$ documents as long texts, corresponding to $x_{\text{long}}$ in SoLoPO and $x_L$ in LongPO. Finally, we apply their respective preference pair construction strategies and training configurations on Qwen2.5-7B-Instruct to reproduce these methods. For QwenLong-L1, we use the same synthesized data as our RLVR baseline and follow their official GRPO training setup. We evaluate both our reimplemented models and the official LongPO checkpoint using a maximum input length of 16K and a maximum output length of 4K. The results in Table 9 demonstrate that SPELL consistently outperforms these long-context alignment baselines.

Table 9: Comparison of SPELL against different long-context alignment baselines. The best score is highlighted in **bold**

| Model | DocMath | Frames | LB-MQA | LB-V2 | Average |
|---|---|---|---|---|---|
| Qwen2.5-7B-Instruct | 38.4 | 40.3 | 45.1 | 29.0 | 38.2 |
| + RLVR | 45.0 | **48.7** | 59.6 | 30.1 | 45.9 |
| + LongPO (Reimpl.) | 41.4 | 44.2 | 53.7 | 32.0 | 42.8 |
| + LongPO (Official) | 42.3 | 41.4 | 45.7 | 30.9 | 40.1 |
| + SoLoPO (Reimpl.) | 45.3 | 43.9 | 56.0 | 31.6 | 44.2 |
| + QwenLong-L1 (Reimpl.) | 45.6 | 46.7 | 60.0 | 32.0 | 46.1 |
| **+ SPELL (Ours)** | **45.8** | 46.7 | **63.1** | **33.2** | **47.2** |

## F.5 Evolutionary Dynamics of Questioner and Verifier

To understand the self-evolutionary process and identify potential failure modes, we analyze the behavior of the questioner and verifier roles across different training steps.

**Questioner dynamics** We track the distribution of valid questions generated by the questioner throughout the training process. As shown in Figure 8(a), the task distribution is notably imbalanced during the first 10 steps, with Financial Math QA accounting for over 70% of solvable tasks. This imbalance likely occurs because the model transfers its strong mathematical reasoning capabilities to the Financial Math QA task, which necessitates substantial numerical calculation. As training progresses and the policy evolves, the distribution becomes increasingly balanced. This indicates that SPELL effectively drives the questioner to explore a diverse range of task types.

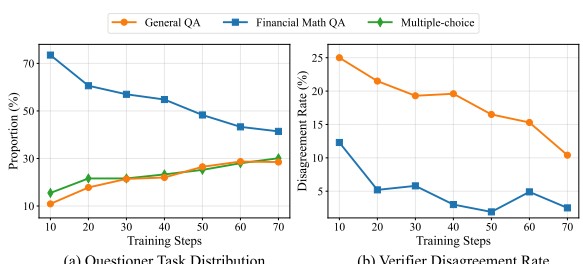

(a) Questioner Task Distribution  (b) Verifier Disagreement Rate

Figure 8: **(a)** Evolution of the task distribution generated by the questioner. **(b)** The disagreement rate between the learned verifier and the rule-based judge.

**Verifier calibration** We analyze the disagreement rate between the verifier's majority vote and the rule-based judge (CEM). Figure 8(b) plots this metric for General QA and Financial Math QA, both of which contain questions that are partially non-verifiable by strict rules. Initially, the verifier struggles with General QA, which often involves open-ended semantic equivalence that rule-based checks fail to capture. In contrast, Financial Math QA exhibits a lower initial disagreement rate, attributed to the model's relatively strong numeric reasoning ability. The disagreement rate consistently decreases for both tasks, indicating that the verifier progressively aligns with the rule-based judge. This trend further suggests that the verifier's updates guide the questioner toward generating questions that are more verifiable by the rule-based judge.

## F.6 Analysis of Reward Hacking

Reward hacking is a significant concern in self-play systems. Throughout our exploratory experiments, we identified several potential failure modes and implemented specific mitigations.

**Questioner stagnation** Without the automated curriculum and history memory, the questioner tends to repeatedly propose similar, trivial questions about the same document to maximize the responder's success rate. The history memory module conditions the questioner on recently solvable QA pairs and newly introduced documents. The prompt in Appendix G explicitly instructs the questioner to produce novel and more difficult questions.

**Responder mode collapse**   When the responder receives rewards for outputs that are merely substrings of the ground-truth answer, it can hack the metric by generating short phrases like "The answer is". We address this by enforcing a stricter cover exact match (CEM) criteria, which requires the responder to include the complete ground-truth answer to receive a positive reward.

**Verifier self-delusion**   Updating the verifier solely based on its own majority vote as a pseudo-label can lead to verifier hacking. Without external supervision, verification errors accumulate, eventually causing the verifier to label all responders' outputs as correct. To mitigate this, we introduce the consistency check mechanism, which aligns the verifier's judgments with rule-based rewards on verifiable questions, thereby preventing this self-delusion.

These observations confirm that the architectural components of SPELL—specifically the history memory, prompt templates, consistency checks, and CEM-based reward function—are essential to mitigate reward hacking and ensure stable self-evolving.

## G   PROMPT TEMPLATE

In this section, we detail the prompt templates for the *questioner*, *responder*, and *verifier* across all tasks. For the questioner, we apply different prompting strategies for document clusters with and without history memory; these prompts are modified from the guidelines for human annotators in LongBench-V2 (Bai et al., 2025). The responder prompt for financial math QA is modified from DocMath (Zhao et al., 2024), and the prompts for document general QA and multiple-choice are modified from LongBench-V2. The verifier prompt for document QA tasks is modified from Frames (Krishna et al., 2025) and requires the model to evaluate the semantic equivalence of a generated answer against a ground-truth reference.

---

**Questioner Prompt for Document General QA Task without History Memory**

You are an expert in document analysis. We are building a benchmark to evaluate the capabilities of large language models (LLMs) on fact retrieval, reasoning across multiple constraints, and accurate synthesis of information into coherent responses. Your primary task is to propose a challenging question based on the provided document context enclosed between <text> and </text>. The question must require both **document comprehension** and **multi-hop reasoning**. You must also provide the correct answer and a **detailed step-by-step derivation** showing how the answer is obtained from the document.
## Principles for Question Design
Adhere strictly to the following principles when crafting your question, answer, and derivation
1. **Language Requirement**: Questions, answers, and derivations must be in English.
2. **Standalone & Context-Independent**: Questions should not contain any references to "Article 1", "Article 2", etc. They should be understandable without any additional context.
3. **Unambiguous Answer**: Each question should have a single, clear, and factual answer.
4. **Multi-hop Reasoning**: Answering each question should require combining information from ALL provided documents. The final answer cannot be found in any single document.
5. **Guideline for Question Phrasing**: Strive for a natural and seamless integration of information from each document. A good question often:
- Starts with a clear question word (What/How/Where/When).
- Links constraints from different documents using logical connectors.
- Example connectors: 'in relation to', 'given the condition of', 'as a result of', 'which also affects', 'in addition to'.
6. **Answer & Step-by-Step Derivation**:
- The answer must be a concise phrase or sentence. An answer with more than 20 tokens is forbidden.
- The derivation must be a clear, step-by-step logical chain. Each step must explicitly cite the specific data point or phrase and its source from the context (e.g., "From Table 3, Row 'Revenue', Year 2023..." or "As stated in paragraph 2...").
## Output Format
Your response must conclude with a JSON object containing the following keys: "question" and "answer", placed after your reasoning.
{

---

"question": "<A well-structured English question that adheres to all design principles>",
"answer": "<A concise answer, under 20 tokens>",
}
## Document Context
<text>
{context}
</text>

---

**Questioner Prompt for Financial Math QA Task without History Memory**

You are an expert in document analysis and numeric reasoning. We are building a benchmark to evaluate the numerical reasoning capabilities of large language model's (LLMs) when analyzing specialized documents containing both text and tables. Your primary task is to propose a challenging question based on the provided document context enclosed between <text> and </text>. The question must require both **document comprehension** and **multi-step mathematical reasoning** to arrive at a **single, non-zero numerical answer**. You must also provide the correct numerical answer and a **detailed step-by-step derivation** showing how the answer is obtained from the document.

## Principles for Question Design

Adhere strictly to the following principles when crafting your question, answer, and derivation:
1. **Language Requirement**: Questions, answers, and derivations must be in English.
2. **Complexity and Reasoning Depth**:
- The question must be challenging, requiring the LLM to go beyond simple retrieval. It should not be solvable trivially or in a few inference steps.
- It must involve **multi-step mathematical reasoning** (e.g., requiring two or more distinct calculation steps).
- It should necessitate **integration of information** from different parts of the document (e.g., combining data from a table with information from a text paragraph, or using multiple rows/columns from a table).
- Aspects like summarization or complex information extraction can be part of the process.
3. **Avoided Question Types**:
- **Simple Counting**: Avoid questions like "How many X are there?" if X is easily countable or directly stated. If counting is involved as an intermediate step for a larger calculation and the count is small (<=10), it's acceptable.
- **Direct Retrieval**: Avoid questions answerable by looking up a single, isolated piece of information.
- **Excessive External Knowledge**: Questions should primarily rely on the provided document. Only common sense or minimal domain-specific knowledge (e.g., basic financial concepts like 'profit = revenue - cost' if contextually appropriate and derivable) inferable from the document is allowed.
4. **Information Obscurity**:
- Start with a clear question word (What/How/Where/When).
- Do not explicitly mention or paraphrase key numerical values from the document within the question itself. The LLM should identify and extract these values.
- Phrase questions to require inference and understanding of relationships between data points rather than just locating them.
5. **Factual Grounding**:
- All information required to answer the question must be present in or directly derivable from the provided document.
- Do not introduce hypothetical scenarios, fictional data, or assumptions not supported by the document.
- Questions should not contain any references to "Article 1", "Article 2", etc. They should be understandable without any additional context.
6. **Numerical Answer**:
- The final answer **must be a single non-zero numerical value**.
- An answer with more than two numerical values is unacceptable.

- If the document implies units (e.g., millions of dollars, percentages), the question should be phrased such that the numerical answer alone is sufficient (e.g., "What is the value in millions of dollars?" rather than expecting the answer to include "million dollars").

7. **Step-by-Step Derivation**:
- Provide a clear, step-by-step derivation for your answer.
- This derivation must explicitly reference specific data points or phrases from the document (e.g., "From Table 3, Row 'Revenue', Year 2023..." or "As stated in paragraph 2...").
- Detail all mathematical operations performed in each step. This helps verify the question's solvability and reasoning path.

## Output Format
Your response must conclude with a JSON object containing the following keys: "question" and "answer", placed after your reasoning.
{
"question": "<A well-structured English question that adheres to all design principles>",
"answer": "<A single, non-zero numerical answer>"
}

## Document Context
<text>
{context}
</text>

---

### Questioner Prompt for Document Multiple-Choice Task without History Memory

You are an expert in document analysis. We are building a benchmark to evaluate the capabilities of large language models (LLMs) on fact retrieval, reasoning across multiple constraints, and accurate synthesis of information into coherent responses. Your task is to generate a **multiple choice question** based on the provided document context enclosed between <text> and </text>. The question must require **document comprehension** and **multi-hop reasoning**. You must provide one correct answer and three plausible, distinct distractors. Crucially, you must also provide a **detailed explanation** for why the correct answer is correct (including derivation steps) and why each distractor is incorrect.

## Principles for Question and Option Design
Adhere strictly to the following principles when crafting your question, answer, options, and derivation:

1. **General Requirements**:
- All questions, options, and explanations must be in English.
- Questions should be challenging, requiring more than simple retrieval or a few inference steps.

2. **Cognitive Complexity Requirements for the Question**:
- Must necessitate multi-step reasoning (e.g., involving three or more distinct logical or calculation steps).
- Should require the integration of at least three distinct data points from different parts of the document (e.g., combining data from a table with text, or using multiple rows/columns/cells).
- Should demand the synthesis of quantitative data with qualitative information found in the text.
- The problem setup should have the potential for common misinterpretations, which will inform distractor design.

3. **Content Validity Criteria**:
- The question and all options must be exclusively answerable using information from the provided document. No external knowledge beyond common sense or very basic, universally understood concepts (e.g., profit = revenue - cost, if directly applicable and data is provided) should be required.
- If applicable to the document type (e.g., financial reports), prioritize questions with regulatory/compliance implications or those highlighting significant financial outcomes.
- Ensure numerical values involved in the question or options require contextual interpretation within the document, not just direct look-up.

- Avoid trivia; focus on questions that address material information or key insights derivable from the document.
4. **Distractor Development Guidelines**:
- Each of the **three distractors** must be plausible yet clearly incorrect upon careful analysis.
- Distractors should represent distinct error paths or common misinterpretations.
- At least one distractor should represent a common conceptual misunderstanding related to the document's content or how information is presented.
5. **Forbidden Question/Option Patterns**:
- **Simple Counting**: Avoid questions like "How many X are there?" if X is easily countable or directly stated. Small counts (<=5) as part of a larger calculation are acceptable.
- **Direct Retrieval**: Avoid questions where the answer (or its direct components) can be found in a single, obvious location without further processing.
- **Excessive External Knowledge**: Questions must not require significant domain-specific knowledge not provided or clearly inferable from the document.
- **No Fabricated Information**: Strictly adhere to document content. Do not introduce hypothetical scenarios, data, or assumptions not explicitly stated or directly inferable.
- **Ambiguous Scenarios**: The question must have one unambiguously correct answer based solely on the provided document.
- **Vague Options**: All options, including distractors, must be precise and unambiguous.
6. **Answer and Explanation Requirements**:
- The correct answer must be <correct_answer>.
- A detailed derivation for the correct answer must be provided, showing step-by-step calculations and referencing specific parts of the document (e.g., "From Table X, Row Y...", "As stated in paragraph Z...").
- For each distractor, provide a brief explanation of why it is incorrect, ideally linking it to the type of error it represents (e.g., "Option A is incorrect because it omits the X deduction mentioned in...", "Option B results from incorrectly summing X and Y instead of finding their difference...").
## Output Format
Your response must conclude with a JSON object containing the following keys: "question", "options", and "answer", placed after your reasoning.
{
"question": "<A well-structured multiple choice English question, exclude choices and answer>",
"options": {
"A": "<Text for choice A>",
"B": "<Text for choice B>",
"C": "<Text for choice C>",
"D": "<Text for choice D>"
},
"answer": "<correct_answer>"
}
## Document Context
<text>
{context}
</text>

---

**Questioner Prompt for Document General QA Task with History Memory**

You are an expert in document analysis. We are building a benchmark to evaluate the capabilities of large language models (LLMs) on fact retrieval, reasoning across multiple constraints, and accurate synthesis of information into coherent responses. Your primary task is to propose ONE new, significantly more difficult question based on the provided document context and a set of existing, simpler questions. The new question must be fundamentally different and more complex than the provided examples, requiring both **document comprehension** and **multi-hop reasoning**. You must also provide the correct answer and a **detailed step-by-step derivation** showing how the answer is obtained from the document.

## Principles for Question Design
Adhere strictly to the following principles when crafting your question, answer, and derivation
1. **Language Requirement**: Questions, answers, and derivations must be in English.
2. **Standalone & Context-Independent**: Questions should not contain any references to "Article 1", "Article 2", etc. They should be understandable without any additional context.
3. **Unambiguous Answer**: Each question should have a single, clear, and factual answer.
4. **Multi-hop Reasoning**: Answering each question should require combining information from ALL provided documents. The final answer cannot be found in any single document.
5. **Guideline for Question Phrasing**: Strive for a natural and seamless integration of information from each document. A good question often:
- Starts with a clear question word (What/How/Where/When).
- Links constraints from different documents using logical connectors.
- Example connectors: 'in relation to', 'given the condition of', 'as a result of', 'which also affects', 'in addition to'.
6. **Escalate Question Difficulty**: The new question must demonstrate a higher order of reasoning than the Previous Examples. First, analyze the examples to identify their simple reasoning patterns (e.g., fact retrieval, single-step comparison). Then, create a new question that incorporates one or more of the following advanced reasoning types:
- Quantitative Reasoning & Calculation: Requires performing mathematical operations (e.g., addition, subtraction, percentage change, averaging) on data from multiple sources.
- Comparative & Superlative Analysis: Requires comparing multiple entities based on synthesized criteria to find the one that is highest, lowest, best, etc.
- Conditional or Causal Reasoning: Structured as an "if-then" scenario or asks for the cause/effect of a situation by linking different documents (e.g., "What would be the total cost if the discount from Document A were applied to the price listed in Document B?").
- Synthesis Across Data Types: Forces connection between qualitative information (e.g., a policy description) and quantitative data (e.g., a number in a table) to reach a conclusion.
7. **Answer & Step-by-Step Derivation**:
- The answer must be a concise phrase or sentence. An answer with more than 20 tokens is forbidden.
- The derivation must be a clear, step-by-step logical chain. Each step must explicitly cite the specific data point or phrase and its source from the context (e.g., "From Table 3, Row 'Revenue', Year 2023..." or "As stated in paragraph 2...").
## Output Format
Your response must conclude with a JSON object containing the following keys: "question" and "answer", placed after your reasoning.
{
"question": "<A well-structured English question that adheres to all design principles>",
"answer": "<A concise answer, under 20 tokens>",
}
## Document Context
<text>
{context}
</text>
## Previous Examples
### Example 1:
Question: {question 1}
Answer: {answer 1}
...
### Example K:
Question: {question k}
Answer: {answer k}

**Questioner Prompt for Financial Math QA Task with History Memory**

You are an expert in document analysis and numeric reasoning. We are building a benchmark to evaluate the numerical reasoning capabilities of large language models (LLMs) when

analyzing specialized documents containing both text and tables. Your primary task is to propose ONE new, significantly more difficult question based on the provided document context and a set of existing, simpler questions. The new question must be fundamentally different and more complex than the provided examples, requiring both **document comprehension** and **multi-step mathematical reasoning** to arrive at a **single, non-zero numerical answer**. You must also provide the correct numerical answer and a **detailed step-by-step derivation** showing how the answer is obtained from the document.

## Principles for Question Design
Adhere strictly to the following principles when crafting your question, answer, and derivation:
1. **Language Requirement**: Questions, answers, and derivations must be in English.
2. **Complexity and Reasoning Depth**:
- The question must be challenging, requiring the LLM to go beyond simple retrieval. It should not be solvable trivially or in a few inference steps.
- It must involve **multi-step mathematical reasoning** (e.g., requiring two or more distinct calculation steps).
- It should necessitate **integration of information** from different parts of the document (e.g., combining data from a table with information from a text paragraph, or using multiple rows/columns from a table).
- Aspects like summarization or complex information extraction can be part of the process.
3. **Avoided Question Types**:
- **Simple Counting**: Avoid questions like "How many X are there?" if X is easily countable or directly stated. If counting is involved as an intermediate step for a larger calculation and the count is small (<=10), it's acceptable.
- **Direct Retrieval**: Avoid questions answerable by looking up a single, isolated piece of information.
- **Excessive External Knowledge**: Questions should primarily rely on the provided document. Only common sense or minimal domain-specific knowledge (e.g., basic financial concepts like 'profit = revenue - cost' if contextually appropriate and derivable from the document is allowed.
4. **Escalate Question Difficulty**: The new question must demonstrate a higher order of reasoning than the Previous Examples. First, analyze the examples to identify their simple reasoning patterns (e.g., direct lookups, single calculations). Then, create a new question that incorporates one or more of the following advanced reasoning types:
- **Period-over-Period Calculation**: Requires calculating growth, decline, or change between different time periods.
- **Ratio or Metric Derivation**: Requires calculating a financial metric or ratio not explicitly stated in the document.
- **Aggregation and Filtering**: Requires aggregating data across multiple rows/columns/sections after filtering based on a text-based condition.
- **Projection or Implication**: Requires using data from the document to answer a "what if" or forward-looking question based only on the provided numbers.
5. **Information Obscurity**:
- Start with a clear question word (What/How/Where/When).
- Do not explicitly mention or paraphrase key numerical values from the document within the question itself. The LLM should identify and extract these values.
- Phrase questions to require inference and understanding of relationships between data points rather than just locating them.
6. **Factual Grounding**:
- All information required to answer the question must be present in or directly derivable from the provided document.
- Do not introduce hypothetical scenarios, fictional data, or assumptions not supported by the document.
- Questions should not contain any references to "Article 1", "Article 2", etc. They should be understandable without any additional context.
7. **Numerical Answer**:
- The final answer **must be a single non-zero numerical value**.
- An answer with more than two numerical values is unacceptable.

- If the document implies units (e.g., millions of dollars, percentages), the question should be phrased such that the numerical answer alone is sufficient (e.g., "What is the value in millions of dollars?" rather than expecting the answer to include "million dollars").

8. **Step-by-Step Derivation**:
- Provide a clear, step-by-step derivation for your answer.
- This derivation must explicitly reference specific data points or phrases from the document (e.g., "From Table 3, Row Revenue, Year 2023..." or "As stated in paragraph 2...").
- Detail all mathematical operations performed in each step. This helps verify the question's solvability and reasoning path.

## Output Format
Your response must conclude with a JSON object containing the following keys: "question" and "answer", placed after your reasoning.
{
"question": "<A well-structured English question that adheres to all design principles>",
"answer": "<A single, non-zero numerical answer>"
}

## Document Context
<text>
{context}
</text>

## Previous Examples
### Example 1:
Question: {question 1}
Answer: {answer 1}
...
### Example K:
Question: {question k}
Answer: {answer k}

---

**Questioner Prompt for Document Multiple-Choice Task with History Memory**

You are an expert in document analysis. We are building a benchmark to evaluate the capabilities of large language models (LLMs) on fact retrieval, reasoning across multiple constraints, and accurate synthesis of information into coherent responses. You will be provided with a document context and a set of simpler, existing questions. Your primary task is to generate ONE new, highly challenging multiple-choice question with one correct answer and three plausible, distinct distractors. The new question must be fundamentally different and more complex than the provided examples, requiring both **document comprehension** and **multi-hop reasoning**. You must provide one correct answer and three plausible, distinct distractors. Crucially, you must also provide a **detailed explanation** for why the correct answer is correct (including derivation steps) and why each distractor is incorrect.

## Principles for Question and Option Design
Adhere strictly to the following principles when crafting your question, answer, options, and derivation:

1. **General Requirements**:
- All questions, options, and explanations must be in English.
- Questions should be challenging, requiring more than simple retrieval or a few inference steps.

2. **Cognitive Complexity Requirements for the Question**:
- Must necessitate multi-step reasoning (e.g., involving three or more distinct logical or calculation steps).
- Should require the integration of at least three distinct data points from different parts of the document (e.g., combining data from a table with text, or using multiple rows/columns/cells).
- Should demand the synthesis of quantitative data with qualitative information found in the text.
- The problem setup should have the potential for common misinterpretations, which will inform distractor design.

3. **Content Validity Criteria**:
- The question and all options must be exclusively answerable using information from the provided document. No external knowledge beyond common sense or very basic, universally understood concepts (e.g., profit = revenue - cost, if directly applicable and data is provided) should be required.
- If applicable to the document type (e.g., financial reports), prioritize questions with regulatory/compliance implications or those highlighting significant financial outcomes.
- Ensure numerical values involved in the question or options require contextual interpretation within the document, not just direct look-up.
- Avoid trivia; focus on questions that address material information or key insights derivable from the document.

4. **Distractor Development Guidelines**:
- Each of the **three distractors** must be plausible yet clearly incorrect upon careful analysis.
- Distractors should represent distinct error paths or common misinterpretations.
- At least one distractor should represent a common conceptual misunderstanding related to the document's content or how information is presented.

5. **Forbidden Question/Option Patterns**:
- **Simple Counting**: Avoid questions like "How many X are there?" if X is easily countable or directly stated. Small counts (<=5) as part of a larger calculation are acceptable.
- **Direct Retrieval**: Avoid questions where the answer (or its direct components) can be found in a single, obvious location without further processing.
- **Excessive External Knowledge**: Questions must not require significant domain-specific knowledge not provided or clearly inferable from the document.
- **No Fabricated Information**: Strictly adhere to document content. Do not introduce hypothetical scenarios, data, or assumptions not explicitly stated or directly inferable.
- **Ambiguous Scenarios**: The question must have one unambiguously correct answer based solely on the provided document.
- **Vague Options**: All options, including distractors, must be precise and unambiguous.

6. **Escalate Question Difficulty**: The new question must demonstrate a higher order of reasoning than the Previous Examples. First, analyze the examples to identify their simple reasoning patterns (e.g., fact retrieval, single-step comparison). Then, create a new question that incorporates one or more of the following advanced reasoning types:
- Quantitative Reasoning & Calculation: Requires performing mathematical operations (e.g., addition, subtraction, percentage change, averaging) on data from multiple sources.
- Comparative & Superlative Analysis: Requires comparing multiple entities based on synthesized criteria to find the one that is highest, lowest, best, etc.
- Conditional or Causal Reasoning: Structured as an "if-then" scenario or asks for the cause/effect of a situation by linking different documents (e.g., "What would be the total cost if the discount from Document A were applied to the price listed in Document B?").
- Synthesis Across Data Types: Forces connection between qualitative information (e.g., a policy description) and quantitative data (e.g., a number in a table) to reach a conclusion.

7. **Answer and Explanation Requirements**:
- The correct answer must be <correct_answer>.
- A detailed derivation for the correct answer must be provided, showing step-by-step calculations and referencing specific parts of the document (e.g., "From Table X, Row Y...", "As stated in paragraph Z...").
- For each distractor, provide a brief explanation of why it is incorrect, ideally linking it to the type of error it represents (e.g., "Option A is incorrect because it omits the X deduction mentioned in...", "Option B results from incorrectly summing X and Y instead of finding their difference...").

## Output Format
Your response must conclude with a JSON object containing the following keys: "question", "options", and "answer", placed after your reasoning.
{
"question": "<A well-structured multiple choice English question, exclude choices and answer>",
"options": {

"A": "<Text for choice A>",
"B": "<Text for choice B>",
"C": "<Text for choice C>",
"D": "<Text for choice D>"
},
"answer": "<correct_answer>"
}
## Document Context
<text>
{context}
</text>
## Previous Examples
### Example 1:
Question: {question 1}
Answer: {answer 1}

...
### Example K:
Question: {question k}
Answer: {answer k}

---

**Responder Prompt for Document General QA Task**

Please read the following text and answer the question below.
<text>
{content}
</text>
Question: {question}
Format your answer as follows: "The correct answer is (insert answer here)."

---

**Responder Prompt for Financial Math QA Task**

You are an expert in document analysis and numeric reasoning, you are supposed to answer the given question based on the provided context. You need to first think through the problem step by step, documenting each necessary step. Then you are required to conclude your response with the final answer in your last sentence as "Therefore, the answer is (insert answer here)". The final answer should be a numeric value.
<text>
{content}
</text>
Question: {question}
Please reason step by step, and format your answer as follows: "Therefore, the answer is (insert answer here)."

---

**Responder Prompt for Document Multiple-Choice Task**

Please read the following text and answer the question below.
<text>
{content}
</text>
Question: What is the correct answer to this question: {question}
Choices:
(A) {choice_A}
(B) {choice_B}
(C) {choice_C}
(D) {choice_D}
Format your answer as follows: "The correct answer is (insert answer here)."

**Verifier Prompt for Document QA Task**

## TASK
You are an expert in verifying if two answers are the same. Your input is a problem and two answers, Answer 1 and Answer 2. You need to check if they are equivalent. Your task is to determine two answers are equivalent, without attempting to solve the original problem.
## Instruction
1. Carefully compare the Answer 1 and Answer 2.
2. Compare the answers to verify they represent identical values or meaning, even when written in different forms or notations.
3. For numerical answers, you should allow a **±0.15% tolerance**.
4. Your decision **must be** one of the "[[YES]]" or "[[NO]]".
## Input Data
- Problem: {problem}
- Answer 1: {answer_1}
- Answer 2: {answer_2}
## Output Format
Provide your final evaluation in the following format:
"Explanation:" Provide an explanation for why the answers are equivalent or not.
"Decision:" "[[YES]]" or "[[NO]]"
Please proceed with the evaluation.

