# OpenReview forum: "SPELL: Self-Play Reinforcement Learning for Evolving Long-Context Language Models"
_ICLR.cc/2026/Conference — ICLR 2026 Poster_

### Official Review · Reviewer_Zsjp · 2025-10-15

**Soundness:** 3
**Presentation:** 3
**Contribution:** 3
**Rating:** 6
**Confidence:** 4

**Summary:**

The paper proposes a self-play RL framework for training LLMs to do long-context reasoning. Since there is little human annotations for long-context reasoning questions, they design a framework which includes a questioner, a responder and a verifier. Each module is trained via RLVR with a specially designed reward function. Overall the experiment results demonstrate that their proposed SPELL method is able to improve the long-context ability of LLMs and surpass the vanilla RLVR.

**Strengths:**

- The problem of long-context reasoning is very relevant to the field. The self-play framework which includes three well-specified roles for generating questions, generating responses and verifying responses is carefully designed with reasonable rewards.
- An interesting design is the iterative curriculum which keeps generating more and more difficult questions. They demonstrate in experiment results that the increasing difficult questions are better than static questions in RLVR.
- Their experiments show that SPELL has consistent advantages over RLVR on most datasets and both long and short context settings. In addition, the verifier and the consistency rewards are useful for the overall model training.

**Weaknesses:**

- The paper aims to improve long-context reasoning ability. The method is quite effective but seems more like a general RLVR training data generation method, and less relevant to the problem of long-context. It would be better to compare their SPELL method with at least one long-context alignment method other than only RLVR. For example, authors mentioned LongPO, SoLoPO and QwenLong-L1, which seems to be good baselines (and can be compared with SPELL given same documents as the starting point).

**Questions:**

- For the questioner reward, why do you use a Gaussian-shaped reward instead of the linear piecewise reward as in R-zero? Is there a specific reason that the Gaussian reward is better? Why do you set σ = 0.5/3 and are there any ablation studies on this?

---

> ### Author Response · Authors · 2025-11-19
> **Response to Reviewer Zsjp (1/2)**
>
> Thank you for your insightful feedback and for acknowledging the strengths of our work, particularly the novelty of the self-play framework and the effectiveness of the iterative curriculum. We appreciate the opportunity to clarify our contributions and address your concerns.
>
> > **Q1: Regarding relevance to the problem of long-context.**
>
> **A1:** Thank you for this critical question. We would like to clarify that **SPELL is a self-play framework tailored for long-context scenarios, rather than a general data generation method.** Its core components are designed to overcome challenges in long-context scenarios:
>
> 1.  **Enhancing grounding in long-context reasoning:** A key challenge in long-context tasks is grounding, i.e., the ability to locate and integrate relevant information scattered across extensive documents while ignoring distractors. SPELL improves grounding by introducing information asymmetry between the questioner and responder. The questioner generates a QA pair based on a small subset of the documents. The responder, however, must answer the question using all documents, which contain distractors. This design forces the responder to develop grounding capabilities, as it must identify the key information within a long, noisy context to match the answer derived from a shorter, focused context.
>
> 2.  **Overcoming unreliable rewards in open-ended QA:** For open-ended long-context questions, answers can be semantically correct yet lexically diverse. This makes simple rule-based rewards like string matching highly unreliable. To address this, we introduce the verifier role, which is trained to assess semantic equivalence. Moreover, to ensure the verifier itself does not become unreliable, we designed a consistency check mechanism that calibrates its majority votes against verifiable rule-based rewards. This provides a robust reward signal for evolving the model's self-judging capability, which is essential for effective self-play on non-verifiable long-context questions.
>
> > **Q2: Regarding more long-context alignment baselines.**
>
> **A2:** We conduct additional experiments to compare SPELL with three long-context alignment baselines: **SoLoPO**, **LongPO**, and **QwenLong-L1**, with Qwen2.5-7B-Instruct as the base model. To ensure a fair comparison, we reimplement these methods using the same raw document corpus as SPELL.
>
> For SoLoPO and LongPO, the core idea is to construct preference pairs for preference optimization training from short contexts containing key information and long contexts containing distractors under the same question. For each data instance with n documents, we first randomly select m=5 documents as short texts and use DeepSeek-R1-0528 to act as the questioner to generate QA pairs. Then, we use DeepSeek-R1-0528 to act as the responder to answer the proposed question using all n documents. We keep the QA pairs where the answers of the questioner and responder are consistent. Next, we take the 5 documents from the questioning stage as short texts, corresponding to $x_\text{short}$ in SoLoPO and $x_S$ in LongPO, and take all n documents as long texts, corresponding to $x_\text{long}$ in SoLoPO and $x_L$ in LongPO. Finally, we follow their preference pair construction strategies and training settings on Qwen2.5-7B-Instruct to reproduce their methods. For QwenLong-L1, we use the same synthesized data as our RLVR baseline and follow their official GRPO training setup. We report results on our reimplemented models as well as the official LongPO checkpoint. The experiments are conducted with a 16K maximum input length and 4K maximum output length.
>
> |Model|DocMath|Frames|LB-MQA|LB-V2|Avg.|
> |:---|:---:|:---:|:---:|:---:|:---:|
> |Qwen2.5-7B-Instruct|38.4|40.3|45.1|29.0|38.2|
> |+RLVR|45.0|48.7|59.6|30.1|45.9|
> |+LongPO (Reimplemented)|41.4|44.2|53.7|32.0|42.8|
> |+LongPO (Official Ckpt)|42.3|41.4|45.7|30.9|40.1|
> |+SoLoPO (Reimplemented)|45.3|43.9|56.0|31.6|44.2|
> |+QwenLong-L1 (Reimplemented)|45.6|46.7|60.0|32.0|46.1|
> |**+SPELL (Ours)**|45.8|46.7|63.1|33.2|**47.2**|
>
> The results demonstrate that **SPELL consistently outperforms these long-context alignment baselines**.

---

> ### Author Response · Authors · 2025-11-19
> **Response to Reviewer Zsjp (2/2)**
>
> > **Q3: Regarding the questioner reward design.**
>
> **A3:** Thank you for these excellent questions about our reward design. We address them in three parts:
>
> 1.  **Gaussian vs. linear piecewise reward:** As discussed in Section 4.4 and visualized in Figure 5(a), our **Gaussian reward offers a more targeted incentive compared to the linear piecewise function in R-Zero**. The Gaussian function concentrates the reward sharply around the point of maximal learning efficiency ($\bar{r}^{\text{res}}=0.5$) and imposes a steeper penalty on questions that are either too easy or too hard. This encourages the questioner to focus more precisely on generating samples at the frontier of the responder's competence. Our empirical results in Figure 5 support this, showing that our method leads to more stable training entropy (Figure 5(b)) and more controlled growth in response length (Figure 5(c)), ultimately achieving superior performance (Figure 5(d)).
>
> 2.  **Principled choice of $\sigma = 0.5/3$:** The choice of $\sigma$ is derived from the **statistical properties of the normal distribution**, where approximately 99.7% of data points fall within three standard deviations ($3\sigma$) of the mean. In SPELL, our goal is to concentrate the questioner's reward at the point of the responder's maximal learning efficiency, where $\bar{r}^\text{res} = 0.5$. So the mean $\mu$ is set to 0.5. Given this mean, the distance to either boundary of the valid average reward range [0, 1] is 0.5. By setting $3\sigma = 0.5$, we ensure the effective range of the questioner reward covers the responder reward space, which yields $\sigma = 0.5/3$.
>
> 3.  **Ablation Study on $\sigma$:** To further validate this theoretical choice, we conduct an ablation study on the value of $\sigma$ using Qwen2.5-7B-Instruct. We test both smaller ($\sigma = 0.5/6$) and larger ($\sigma = 0.5/2$) values.
>
> |$\sigma$|DocMath|Frames|LB-MQA|LB-V2|Avg.|
> |:---:|:---:|:---:|:---:|:---:|:---:|
> |0.5/6|45.3|**47.0**|62.3|32.8|46.8|
> |0.5/3|**45.8**|46.7|**63.1**|**33.2**|**47.2**|
> |0.5/2|45.5|46.0|59.4|31.8|45.7|
>
> The results show that narrowing the curve ($\sigma = 0.5/6$) has a minimal negative impact, as the reward is still well-focused. However, widening the curve ($\sigma = 0.5/2$) significantly degrades performance, as it assigns more rewards to overly easy or hard questions and provides a less targeted signal. This confirms that $\sigma = 0.5/3$ is **not only theoretically sound but also empirically optimal**.
>
> We hope these clarifications and additional results have addressed your concerns. We will incorporate these discussions and findings into the revised manuscript.

---

> > ### Comment · Reviewer_Zsjp · 2025-11-19
> >
> > Thanks for the authors' reply, which have addressed my concerns. I have raised my score.

---

### Official Review · Reviewer_czEk · 2025-10-29

**Soundness:** 3
**Presentation:** 3
**Contribution:** 3
**Rating:** 6
**Confidence:** 3

**Summary:**

This paper introduces a SPELL (Self-Play Reinforcement Learning for Evolving Long-Context Language Models), a self-play RL framework for long-context reasoning, in which a unified policy is trained using reward signals derived from three roles: the questioner, the responder, and the verifier. The reward functions involve no human supervision and drive self-improvement via an easy-to-hard curriculum. The questioner generates ground-truth answers for the responder and is incentivized to produce questions of intermediate difficulty. The verifier is trained via self-consistency (e.g., agreement across multiple samples). The responder is optimized to match the ground-truth answer and to satisfy the verifier’s majority vote.

SPELL yields consistent gains over a strong RLVR (reinforcement learning with verifiable rewards) baseline across 12 open-source models on six long-context QA benchmarks, evaluated at maximum input lengths of 16K and 100K tokens. SPELL also generalizes to longer contexts and improves test-time exploration and performance.

**Strengths:**

The paper’s originality lies in extending the single-policy self-play paradigm to long-context understanding and reasoning through a unified tri-role self-play formulation:

(1) A single policy is supervised under three distinct roles, which form a self-improvement loop.

(2) All reward signals are generated entirely through the model’s own self-evaluation, without any human knowledge or supervision. This design fully leverages the base model’s inherent potential to improve and evolve autonomously.

The experimental comparison with RLVR is solid, and the core components of the questioner and verifier are thoroughly examined through ablation studies.

The paper is clearly presented and easy to follow.

**Weaknesses:**

**Baseline breadth.** A major weakness is the absence of comparisons against recent long-context SOTA methods. The main RL baseline is RLVR on static data synthesized by DeepSeek-R1-0528. Adding head-to-head comparisons with long-context SOTA methods would better position SPELL among competitive approaches. In particular, SoLoPO (short-to-long preference optimization) and LongPO (long-context self-evolution via preference optimization) report substantial gains and improved length/domain generalization on long-context benchmarks.

**Risk of self-reinforcing errors.** Because the self-consistency and mutual dependencies among the questioner, responder, and verifier drive the co-evolution, one potential weakness of the SPELL framework is its susceptibility to self-reinforcing errors over time. It would strengthen the work to discuss or incorporate mechanisms that mitigate it.

**Questions:**

1. How often do the verifier and the rule-based judge disagree? The paper mentions that “the CEM-based reward function is brittle and can penalize semantically correct but lexically different answers”. What happens if we replace CEM with a stronger external judge (e.g., an independent 120B model)? In that setting, what is the relative importance of the verifier versus the rule-based judge?

2. How sensitive is performance to G (number of responder and verifier rollouts)?

---

> ### Author Response · Authors · 2025-11-19
> **Response to Reviewer czEk (1/2)**
>
> Thank you for your constructive feedback and for recognizing the originality of our work and the rigor of our experiments. We appreciate the opportunity to address your questions and provide further clarification.
>
> > **Q1: Regarding comparison with more long-context baselines.**
>
> **A1:** We agree that comparing SPELL with other SOTA long-context methods is crucial. We have conducted additional experiments on Qwen2.5-7B-Instruct to compare our method against **SoLoPO** and **LongPO**. To ensure a fair comparison, we reimplement these two methods using the same raw document corpus as SPELL. To construct short texts containing key information and long texts containing distractors for the same question, we first use DeepSeek-R1-0528 to perform data synthesis. Specifically, for each piece of data containing n documents, we first randomly select m=5 documents as short texts and employ DeepSeek-R1-0528 as the questioner to generate question-answer pairs. Then, we employ DeepSeek-R1-0528 as the responder to answer the proposed question using all n documents. We keep the QA pairs where the answers of the questioner and responder are consistent. Next, we use the 5 documents from the questioning stage as short texts, corresponding to $x_\text{short}$ in SoLoPO and $x_S$ in LongPO, and the complete n documents as long texts, corresponding to $x_\text{long}$ in SoLoPO and $x_L$ in LongPO. Finally, we follow their preference pair construction strategies and training settings on Qwen2.5-7B-Instruct to reproduce their methods. We report results on our reimplemented models as well as the official LongPO checkpoint. The experiments are conducted with a 16K maximum input length and a 4K maximum output length.
>
>
> |Model|DocMath|Frames|LB-MQA|LB-V2|Avg.|
> |:---|:---:|:---:|:---:|:---:|:---:|
> |Qwen2.5-7B-Instruct|38.4|40.3|45.1|29.0|38.2|
> |+RLVR|45.0|48.7|59.6|30.1|45.9|
> |+LongPO (Reimplemented)|41.4|44.2|53.7|32.0|42.8|
> |+LongPO (Official Ckpt)|42.3|41.4|45.7|30.9|40.1|
> |+SoLoPO (Reimplemented)|45.3|43.9|56.0|31.6|44.2|
> |**+SPELL (Ours)**|45.8|46.7|63.1|33.2|**47.2**|
>
> The results show that **SPELL consistently outperforms these long-context alignment baselines**. We will include these additional experiments in the revised version of the paper.
>
> > **Q2: Regarding mitigating the risk of self-reinforcing errors.**
>
> **A2:** This is a critical point. We have incorporated several key mechanisms to mitigate the accumulation of self-reinforcing errors in our self-play framework:
>
> -   **Grounding filter:** Before a question is used for training the responder, we verify that it cannot be answered without the documents. This penalizes the questioner ($r^\text{que} = -0.5$) for generating questions that rely on parametric knowledge or hallucination, preventing the model from collapsing into ungrounded reasoning.
>
> -   **Information asymmetry between questioning and responding:**  The questioner generates questions based on a subset of documents (m out of n). This focused context makes the generation of the reference answer relatively easy and reliable. The responder, however, must solve these questions given the full set of n documents, where n-m documents unseen by the questioner act as distractors. We only keep QA pairs where the responder successfully reaches the reference answer, thereby filtering out ambiguous or incorrect questions.
>
>   -  **Gaussian-shaped reward filtering:** Incorrect QA pairs naturally fail to produce the optimal learning signal. Our Gaussian reward function for the questioner centers on a responder success rate of 0.5. Questions with incorrect answers typically result in either near-zero success rates (unsolvable or random guessing) or near-one success rates (consistently matching an incorrect reference), both of which fall into the low-reward regions of the Gaussian curve and are effectively suppressed by the policy optimization process.
>
> -   **Verifier consistency check:** We observe in our early experiments that a verifier trained only on its own self-consistency signals could quickly start labeling all its outputs as correct, a form of self-hacking. To prevent this, we introduce a consistency check, which calibrates the verifier's judgments against a rule-based reward (cover exact match) on verifiable tasks. This prevents the verifier from propagating its own errors to other roles.
>
> Collectively, these mechanisms create a robust self-play system that actively mitigates the risk of self-reinforcing errors. We will add a discussion of these strategies to the revised manuscript.

---

> ### Author Response · Authors · 2025-11-19
> **Response to Reviewer czEk (2/2)**
>
> > **Q3: Regarding disagreement of the verifier and the rule-based judge.**
>
> **A3:** We track the disagreement rate between the verifier's majority vote and the rule-based (CEM) judge every 10 training steps on Qwen2.5-7B-Instruct.
>
> |Step|10|20|30|40|50|60|70|
> |:--:|:--:|:--:|:--:|:--:|:--:|:--:|:--:|
> |Disagreement Rate (%)|21.8|11.8|7.4|6.9|5.9|5.3|3.9|
>
> **Initially, the disagreement is high (21.8%), but it steadily decreases as training progresses.** This indicates that the verifier successfully aligns with the rule-based signal. This trend also suggests that the **verifier's updates guide the questioner to generate questions that are more verifiable by the rule-based check.**
>
> > **Q4: Regarding using a stronger external judge:**
>
> **A4:** We replace the rule-based judge with a stronger external model, gpt-oss-120b, and run the experiments with Qwen2.5-7B-Instruct as the base model.
>
> |Model|DocMath|Frames|LB-MQA|LB-V2|Avg.|
> |:---|:---:|:---:|:---:|:---:|:---:|
> |Qwen2.5-7B-Instruct|38.4|40.3|45.1|29.0|38.2|
> |+SPELL (rule-based judge)|45.8|46.7|**63.1**|**33.2**|**47.2**|
> |+SPELL (gpt-oss-120b judge)|**47.1**|**48.0**|61.6|32.1|**47.2**|
> |+SPELL (rule-based judge) w/o verifier|39.4|46.6|60.4|29.4|44.0|
> |+SPELL (gpt-oss-120b judge) w/o verifier|47.0|47.2|61.1|31.3|46.7|
>
> Our results show that **replacing the rule-based judge with a stronger external judge model does not yield a significant overall improvement**. This suggests Qwen2.5-7B-Instruct is already capable of learning the semantic verification task effectively through self-play, without needing a stronger external supervisor. **When using the stronger judge, the internal verifier's role becomes less critical.** Removing it only causes a minor 0.5-point drop, compared to the larger 3.2-point drop when using the rule-based judge. The insights gained from these experiments and analyses will be incorporated into the updated manuscript.
>
> > **Q5: Regarding sensitivity to G (number of rollouts).**
>
> **A5:** We perform an ablation study on G using Qwen2.5-7B-Instruct.
>
> |Model|DocMath|Frames|LB-MQA|LB-V2|Avg.|
> |:---|:---:|:---:|:---:|:---:|:---:|
> |Qwen2.5-7B-Instruct|38.4|40.3|45.1|29.0|38.2|
> |+RLVR|45.0|**48.7**|59.6|30.1|45.9|
> |+SPELL (G=4)|44.3|47.1|62.5|31.8|46.4|
> |+SPELL (G=8)|45.8|46.7|**63.1**|**33.2**|**47.2**|
> |+SPELL (G=16)|**46.0**|47.4|62.7|31.9|47.0|
>
> The results show that G=8 achieves the best performance. However, G=4 and G=16 also yield strong results, which demonstrates that **SPELL is not sensitive to the number of rollouts**. G=8 represents the best balance between performance and computational cost. We will include this additional ablation experiment and analysis in the revised version of the paper.

---

### Official Review · Reviewer_HDua · 2025-11-01

**Soundness:** 3
**Presentation:** 3
**Contribution:** 2
**Rating:** 4
**Confidence:** 3

**Summary:**

This paper proposes SPELL (Self-Play Reinforcement Learning for Evolving Long-Context Language Models), a framework designed to address the scarcity of human annotations and verifiable rewards in long-context reasoning. The core idea is a multi-role self-play loop where a single LLM policy alternates between three roles:

Questioner ($\pi^{que}_\theta$): Generates question-answer pairs from raw documents, conditioned on a growing history memory to increase difficulty.

Responder ($\pi^{res}_\theta$): Answers the generated questions based on the full document context.

Verifier ($\pi^{ver}_\theta$): Assesses the semantic equivalence between the responder's output and the reference answer via majority voting over multiple rollouts.

**Strengths:**

Well-Motivated Problem: Tackles the critical and underexplored challenge of applying RL to long-context reasoning without reliable human annotations or programmatic verifiers.

Comprehensive Evaluation: The scale of the experimentation is impressive, spanning 12 models of various sizes and architectures across 6 benchmarks. This strongly supports the claim of generalizability.

Strong Ablations: The ablation studies are a key strength, providing convincing evidence for the necessity of each core component (history memory, verifier updates, Gaussian reward).

Effective Methodological Design: The three-role framework is elegant. The Gaussian reward for targeting the "competence frontier" and the hybrid responder reward are intelligent designs that directly address stability and learning efficiency in self-play.

Compelling Results vs. Baselines: Demonstrating superiority over a strong RLVR baseline and instruction-tuned models effectively highlights the value of the adaptive, self-play curriculum.

**Weaknesses:**

Limited Analysis of Scaling Laws: The paper shows results up to 32B models. A key question for a "scalable" method is its behavior on larger, state-of-the-art models (e.g., 70B+). Does the performance gap over baselines widen or narrow?

Superficial Treatment of Computational Cost: The framework requires $G + G^2$ generations per data point, which is computationally expensive. A discussion of training cost (e.g., GPU hours compared to the RLVR baseline) and its trade-off with performance is missing but crucial for a complete assessment.

Potential for "Covariate Shift" in Verification: The verifier is trained and evaluated on data generated by the same, evolving policy. There is a risk of the verifier overfitting to the specific style or errors of its companion roles, leading to a degenerate equilibrium. The paper does not investigate this potential pathology.

Narrow Benchmarking: While the benchmarks are diverse, they are all QA-style tasks. It is unclear if the improvements translate to other long-context tasks like summarization, retrieval-augmented generation, or long-form dialogue.

**Questions:**

1. Computational Overhead: Could the authors provide an estimate of the computational cost (e.g., FLOPs or GPU hours) of SPELL compared to the RLVR baseline? Given the $O(G^2)$ verifier rollouts, how do you justify the cost-benefit trade-off, and are there strategies to reduce this overhead?

2. Verifier Degeneration: The verifier's training signal is based on self-consistency and alignment with a rule-based check. How does the authors ensure the verifier does not degenerate, for example, by developing a biased "style preference" aligned with the responder instead of true semantic equivalence, especially on non-verifiable tasks?

3. Scaling to Larger Models: The results are compelling on models up to 32B. What is your hypothesis regarding how SPELL would perform on much larger foundation models (e.g., 70B or 140B parameters)? Would the adaptive curriculum provide even greater relative gains, or would the need for high-quality data diminish its advantage?

4. Comparison to Simpler Baselines: Did you consider comparing against a simpler "self-training" baseline, where you generate synthetic QA pairs with a teacher model (like DeepSeek-R1) once and then fine-tune the student model on this static dataset? This would help isolate the benefit of the online, adaptive self-play loop from the mere use of synthetic data.

5. Failure Mode Analysis: Beyond aggregate metrics, can you provide a qualitative analysis of common failure cases? For instance, what types of questions does the evolved questioner struggle to generate, and what mistakes does the verifier consistently make? This would greatly inform future iterations of the framework.

---

> ### Author Response · Authors · 2025-11-19
> **Response to  Reviewer HDua (1/2)**
>
> Thank you for your detailed and insightful review. We appreciate your recognition of our work's motivation, comprehensive evaluation, methodological design, and compelling results. We are grateful for the opportunity to address your questions and further clarify our contributions.
>
> > **Q1: Regarding computational overhead.**
>
> **A1:** Thank you for raising this important point. We rerun our experiments on a node equipped with $8 \times$ NVIDIA H20 GPUs for both SPELL and the RLVR baseline, using a batch size of 128 and a group size G=8. The table below details the average time per training step.
>
> |Method|Policy Rollout (s)|Policy Update (s)|Total (s)|
> |:----:|:---|:---:|:---:|
> |RLVR|231.5 (Responder)|388.3|619.8|
> |SPELL|38.3 (Questioner)+186.0 (Responder)+64.2 (Verifier)=288.5|407.5|696.0|
>
> While SPELL involves more rollouts, **the total training time per step is only a moderate 12.4% longer than the RLVR baseline**. Crucially, the RLVR baseline requires using DeepSeek-R1-0528 for offline data synthesis. This offline synthesis incurs a substantial cost that can exceed the GPU hours required for the additional rollouts in SPELL.
>
> Regarding the verifier, its rollouts account for only 22.3% of the total rollout time and 9.2% of the total step time. As shown in our ablation study (Table 2, Lines 429-431), the verifier provides a significant 3.2-point performance gain (from 44.0 to 47.2). We argue that **this performance gain far outweighs the minor increase in training cost**. To minimize this cost, we employ two strategies. First, we don't provide the lengthy documents for the verifier. Second, we extract only the final answer from the responder's full output before passing it to the verifier. Both strategies can reduce the input length and computational overhead for verification.
>
> We will include this detailed cost analysis in the revised paper.
>
> > **Q2: Regarding the potential for verifier degeneration.**
>
> **A2:** We indeed observed verifier degeneration in our early experiments with Qwen2.5-7B-Instruct. When the verifier is updated solely based on its own majority vote as a pseudo-label, its errors are amplified in a self-reinforcing loop. This eventually caused the model to label all of its own outputs as correct. To prevent this, we introduce the **consistency check mechanism** (Line 249-250). This mechanism anchors the verifier's learning to an objective ground truth on verifiable questions. By enforcing agreement with a rule-based judge (cover exact match), we ensure the reliability of the training signal and prevent the verifier from degenerating.
>
> To further validate this mechanism, we track the disagreement rate between the verifier's majority vote and the rule-based judge every 10 training steps on Qwen2.5-7B-Instruct.
> As shown below, the disagreement rate steadily decreases to 3.9%, which confirms that the verifier aligns with the objective rule-based judge rather than the responder's output styles.
>
> |Step|10|20|30|40|50|60|70|
> |:--:|:--:|:--:|:--:|:--:|:--:|:--:|:--:|
> |Disagreement Rate (%)|21.8|11.8|7.4|6.9|5.9|5.3|3.9|
>
> > **Q3: Regarding scaling to larger models.**
>
> **A3:** We hypothesize that **SPELL's advantages over the RLVR baseline will not only persist but likely widen on larger models**. Three key observations support this hypothesis:
>
> 1.  **Positive scaling trend:** Our main experiments on Qwen2.5 models from 7B to 32B already show a clear scaling trend. As shown below, the performance gap between SPELL and RLVR widens as model size increases, suggesting strong potential for scaling further.
>
> |Model|Base|+RLVR|+SPELL|Gap (SPELL vs RLVR)|
> |:---|:--:|:---:|:---:|:---:|
> |Qwen2.5-7B-Instruct|38.2|45.9 (+7.7)|47.2 (+9.0)|+1.3|
> |Qwen2.5-14B-Instruct|50.8|55.8 (+5.0)|57.8 (+7.0)|+2.0|
> |Qwen2.5-32B-Instruct|51.8|56.8 (+5.0)|59.5 (+7.7)|+2.7|
>
> 2.  **Effectiveness on strong models:** As discussed in our Section 4.2 (Lines 314-322 & 370-372), SPELL achieves significant gains even on Qwen3-30B-A3B-Thinking, a strong reasoning model where RLVR yields no improvement. Moreover, our test-time scaling analysis (Figure 3 & Lines 391-398) shows that SPELL improves the pass@8 score by 7.6 points over Qwen3-30B-A3B-Thinking and 6.4 points over RLVR. These findings demonstrate SPELL's ability to elevate the performance ceiling of even highly capable models.
>
> 3.  **Adaptive curriculum:** A key advantage of SPELL is that the training data quality scales with the model. As shown in Figure 4 (left), our curriculum automatically adapts question difficulty to the model's evolving capabilities. Unlike static datasets, which may become too easy for large models, SPELL’s self-adaptive nature ensures the training tasks remain challenging.

---

> ### Author Response · Authors · 2025-11-19
> **Response to  Reviewer HDua (2/2)**
>
> > **Q4: Regarding comparison to simpler self-training baselines.**
>
> **A4:** This is a great suggestion. In fact, **our RLVR baseline is implemented exactly as the self-training approach you described.** We first use a powerful teacher model (DeepSeek-R1-0528) to generate a static dataset of synthetic QA pairs, and then fine-tune the policy model using RL on this dataset. The results in Table 1 and our analysis (Line 314-322 & 370-372) show that **SPELL's online, adaptive self-play loop outperforms this static self-training approach**.
>
> > **Q5: Regarding narrow benchmarking.**
>
> **A5:** We agree that validating our method on a broader range of long-context tasks is essential. To address this, we conduct additional evaluations on two benchmarks:
>
> - **MRCR [1-2]:** A multi-needle "Needle in a Haystack" benchmark.
>
> - **HELMET [3]:** Using three subsets covering Retrieval-Augmented Generation (RAG), Many-shot In-Context Learning (ICL), and Summarization (Summ).
>
> We evaluate the Qwen2.5 base models (7B, 14B, and 32B) against the RLVR baseline and SPELL with a maximum input length of 16K.
>
> |Model|MRCR-2needle|MRCR-4needle|MRCR-8needle|Helmet-RAG|Helmet-ICL|Helmet-Summ|Average|
> |:---|:---:|:---:|:---:|:---:|:---:|:---:|:---:|
> |Qwen2.5-7B|6.9|2.0|2.2|50.0|3.5|4.1|11.4|
> |+RLVR|22.0|12.5|10.5|49.3|2.6|14.3|18.5|
> |**+SPELL**|34.5|16.5|16.0|54.2|10.4|13.7|**24.2**|
> |Qwen2.5-14B|23.1|10.5|10.0|42.4|1.5|3.7|15.2|
> |+RLVR|20.9|9.5|9.1|46.7|42.0|24.7|25.5|
> |**+SPELL**|35.0|22.1|17.8|52.3|39.2|23.0|**31.6**|
> |Qwen2.5-32B|27.0|11.5|12.0|59.0|42.8|16.2|28.1|
> |+RLVR|36.7|14.6|13.0|52.7|16.7|21.0|25.8|
> |**+SPELL**|38.0|18.5|14.7|61.4|56.4|21.2|**35.0**|
>
> The empirical results show that **SPELL consistently and significantly outperforms both the base models and the RLVR baseline across diverse long-context tasks**. We will incorporate these additional experiments into the revised version of the paper.
>
> [1] Michelangelo: Long context evaluations beyond haystacks via latent structure queries. (Vodrahalli, Kiran, et al., arxiv 2024)
>
> [2] Introducing GPT-4.1 in the API. (OpenAI, https://openai.com/index/gpt-4-1/)
>
> [3] HELMET: How to Evaluate Long-Context Language Models Effectively and Thoroughly. (Yen, Howard, et al., ICLR 2025)
>
> > **Q6: Regarding failure mode analysis.**
>
> **A6:** This is a valuable suggestion. To provide deeper insight into SPELL's behavior, we conduct a qualitative analysis focusing on training dynamics and tasks where the model faces challenges.
>
> - **Questioner dynamics:** We analyze the distribution of solvable questions generated by the questioner over training steps. As shown in the table below, the model initially struggles to generate valid General QA and Multiple Choice tasks. As training progresses, it becomes proficient at these tasks, eventually achieving a balanced distribution of question types.
>
> |Training Steps|General QA|Multiple Choice|Financial Math QA|
> |:---:|:---:|:---:|:---:|
> |1–10|10.9%|15.5%|73.5%|
> |11–20|17.8%|21.6%|60.6%|
> |21–30|21.4%|21.6%|57.0%|
> |31–40|22.0%|23.3%|54.8%|
> |41–50|26.5%|25.2%|48.3%|
> |51–60|28.7%|28.0%|43.3%|
> |61–70|28.5%|30.1%|41.4%|
>
> - **Verifier challenges:** We analyze the verifier's disagreement rate with the rule-based judge for Financial Math QA and General QA. These tasks contain questions that are partially non-verifiable.
>
> |Training Steps|General QA|Financial Math QA|
> |:---:|:---:|:---:|
> |1–10|25.0%|12.3%|
> |11–20|21.5%|5.2%|
> |21–30|19.3%|5.8%|
> |31–40|19.6%|3.0%|
> |41–50|16.5%|1.9%|
> |51–60|15.3%|4.9%|
> |61–70|10.4%|2.5%|
>
> The verifier initially struggles with the open-ended General QA task, which requires semantic judgment. While the disagreement rate decreases for both tasks over time, a gap remains in General QA. This highlights that calibrating the verifier's judgment on non-verifiable, open-ended tasks remains a key challenge and an important direction for future work. We will include this analysis in our revised manuscript.

---

> ### Comment · Reviewer_HDua · 2025-11-20
>
> I thank the authors for their detailed and comprehensive response. It has adequately addressed most of my concerns. Accordingly, I have decided to raise my score for this paper.

---

### Official Review · Reviewer_wF6e · 2025-11-01

**Soundness:** 4
**Presentation:** 3
**Contribution:** 4
**Rating:** 8
**Confidence:** 3

**Summary:**

The paper propose SPELL, a multi-role self-play reinforcement learning framework that enables scalable, label-free optimization for long-context reasoning. SPELL integrates three cyclical roles—questioner, responder, and verifier—within a single model to enable continual self-improvement. Extensive experiments on six long-context benchmarks show that SPELL consistently improves performance across diverse LLMs and outperforms equally sized models fine-tuned on large-scale annotated data.

**Strengths:**

1. The motivation of this paper is well-grounded.
2. The multi-agent RL system designed is both novel and sound. The reward designed is intuitively justified and appears effective for long-context reasoning.
3. The experiments is extensive and carefully designed.

**Weaknesses:**

Given the unreliability of human annotations in long-context reasoning, how do you ensure that the questions and their corresponding answers provided by the questioner are correct?

**Questions:**

Given that the entire multi-agent RL pipeline is largely hand-engineered, the paper should discuss whether reward hacking could theoretically occur. For example, the response generator might collapse to a short, low-effort policy while the questioner learns to ask only trivial questions, yet the pair still receives a high combined reward.

---

> ### Author Response · Authors · 2025-11-19
> **Response to Reviewer wF6e**
>
> Thank you for your thorough review and for recognizing the novelty, soundness, and extensive experimental rigor of our work. Below, we address your questions regarding data reliability and potential reward hacking.
>
> > **Q1: Regarding the correctness of self-generated questions and answers.**
>
> **A1:** This is a critical question for any self-play framework. In our context, a correct QA pair is one where the question is grounded in the document and the reference answer can be logically derived from it. To ensure reliability without human labels, SPELL integrates four verification mechanisms:
>
> -   **Grounding filter:** After a question is generated, the responder attempts to answer it without access to the documents. If the model answers correctly, the question likely relies on parametric knowledge rather than the provided context. Such questions are discarded, and the questioner is penalized ($r^{\text{que}} = -0.5$) to discourage the generation of non-grounded or hallucinated questions.
>
> -   **Information asymmetry between questioning and responding:**  The questioner generates questions based on a subset of documents (m out of n). This focused context makes the generation of the reference answer relatively easy and reliable. The responder, however, must solve these questions given the full set of n documents, where n-m documents unseen by the questioner act as distractors. We only keep QA pairs where the responder successfully reaches the reference answer, thereby filtering out ambiguous or incorrect questions.
>
> -  **Gaussian-shaped reward filtering:** Incorrect QA pairs naturally fail to produce the optimal learning signal. Our Gaussian reward function for the questioner centers on a responder success rate of 0.5. Questions with incorrect answers typically result in either near-zero success rates (unsolvable or random guessing) or near-one success rates (consistently matching an incorrect reference), both of which fall into the low-reward regions of the Gaussian curve and are effectively suppressed by the policy optimization process.
>
> - **Verifier calibration via rule-based anchors:** The verifier is anchored by a rule-based judge (cover exact match). This ensures the verifier learns a correct judgment signal on verifiable questions. A calibrated verifier then provides reliable feedback for the other roles, which not only prevents the responder from being rewarded for incorrect answers but also prevents the questioner from receiving false positive feedback for generating incorrect questions.
>
> We will include this detailed discussion on data reliability mechanisms in the revised manuscript.
>
> > **Q2: Regarding the potential for reward hacking.**
>
> **A2:** We appreciate this insightful theoretical consideration. Reward hacking is a concern in self-play systems. **The specific trivial loop you described can be effectively prevented by our Gaussian-shaped reward function**. The questioner receives a near-zero reward when the question is too easy, which results in a negative advantage (Eq. 9) relative to other questions in the batch. This forces the questioner to generate questions of medium difficulty to maximize its reward, thus breaking the trivial loop. Beyond this specific case, we did observe and mitigate some forms of reward hacking during our exploratory experiments:
>
> -   **Verifier self-delusion:** Updating the verifier solely based on its own majority vote as a pseudo-label can lead to verifier hacking. During the self-training loop, the verifier's errors are accumulated, which ultimately leads it to label all its own outputs as correct. To solve this, we introduce the **consistency check** mechanism (lines 249-250) for the verifier. This mechanism prevents the verifier from hacking itself by aligning it with rule-based rewards on verifiable questions.
>
> -   **Questioner stagnation:** Without the **automated curriculum and history memory** (Section 3.1), we observed the questioner repeatedly proposing similar questions about the same document. The history memory ensures that when generating a new question, the questioner is conditioned on recently solved QA pairs and new documents. The prompt (Appendix H, Lines 1286-1530) explicitly guides it to generate novel and more difficult questions.
>
> -   **Responder mode collapse:** If the responder could also get a reward when its response is a substring of the ground-truth answer, it quickly learned to hack this by only outputting short, useless phrases like "The answer is". We addressed this by making the reward stricter: the responder must include the complete ground-truth answer to get a reward.
>
> These examples show that the hand-engineered components of SPELL are specifically designed to mitigate the risk of reward hacking. We will add a discussion of these potential failure modes and our solutions in the revised manuscript.

---

### Meta-Review · Area_Chair_M9ru · 2026-01-06

**Summary:**

The paper proposed SPELL, a multi-role self-play RL framework where a single LLM alternates between questioner, responder, and verifier roles to enable label-free optimization for long-context reasoning. The primary concerns centered on potential for reward hacking and self-reinforcing errors in the self-play loop; missing comparisons with recent long-context alignment methods like SoLoPO and LongPO; computational overhead from multiple verifier rollouts; narrow benchmarking limited to QA-style tasks; risk of verifier degeneration through overfitting to companion roles; and insufficient justification for design choices such as the Gaussian-shaped questioner reward and rollout group size G.

**Reviewer Concerns:**

The authors addressed most concerns with substantive experiments and analysis.

For reward hacking and self-reinforcing errors, the authors detailed four mitigation mechanisms: grounding filters, information asymmetry, Gaussian-shaped reward filtering, and verifier consistency checks anchored to rule-based judgments.

The concern about scaling to 70B+ models remains unaddressed experimentally, though the authors demonstrated a widening performance gap over RLVR from 7B to 32B models.

**Reviewer Scores:**

Reviewer scores would likely remain unchanged given no substantial outstanding concerns.

---

### Decision · Program_Chairs · 2026-01-26

Accept (Poster)